# Knowing When to Quit: Probabilistic Early Exits for Speech Separation Networks

**Kenny Falkær Olsen**[1,2]   **Mads Østergaard**[2]   **Karl Ulbæk**[2]   **Søren Føns Nielsen**[2]
**Rasmus Malik Høegh Lindrup**[2]   **Bjørn Sand Jensen**[1]   **Morten Mørup**[1]

[1]Technical University of Denmark   [2]WS Audiology

## Abstract

In recent years, deep learning-based single-channel speech separation has improved considerably, in large part driven by increasingly compute- and parameter-efficient neural network architectures. Most such architectures are, however, designed with a fixed compute and parameter budget and consequently cannot scale to varying compute demands or resources, which limits their use in embedded and heterogeneous devices such as mobile phones and hearables. To enable such use-cases we design a neural network architecture for speech separation and enhancement capable of early-exit, and we propose an uncertainty-aware probabilistic framework to jointly model the clean speech signal and error variance which we use to derive probabilistic early-exit conditions in terms of desired signal-to-noise ratios. We evaluate our methods on both speech separation and enhancement tasks where we demonstrate that early-exit capabilities can be introduced without compromising reconstruction, and that when trained on variable-length audio our early-exit conditions are well-calibrated and lead to considerable compute savings when used to dynamically scale compute at test time while remaining directly interpretable.

## 1 Introduction

The cocktail party problem (Cherry, 1953) concerns the separation of a (possibly unknown) number of overlapping speakers, potentially corrupted by environmental noise and reverberation. The task is typically divided into speech separation (separating multiple speakers) and speech enhancement (removing environmental noise and/or reverberation), which have many applications in, e.g., telecommunications and hearables.

Single-channel speech separation has become predominantly deep learning-based with the introduction of the Time-domain Audio Separation Network (TasNet) (Luo & Mesgarani, 2018). Notable follow-ups include Conv-TasNet (Luo & Mesgarani, 2019), which demonstrated competitive speech separation performance could be achieved at significantly lower computational cost than previously held, and SepFormer (Subakan et al., 2021), which showed large performance improvements using a transformer-based architecture. While the transformer improved modeling performance and training parallelism over prior recurrent and convolutional networks it also tied computational complexity to the length of the attention context window, which may become prohibitive for long sequence or in online contexts. For training, these works have primarily relied on the scale-invariant signal-to-noise ratio (SI-SNR) loss (Roux et al., 2018) with utterance-level permutation invariant training (uPIT) (Yu et al., 2017; Kolbæk et al., 2017).

Current state-of-the-art (SOTA) speech separation and enhancement architectures typically cannot vary their compute in response to simplifying conditions in the input such as non-overlapping speech, low environmental noise, or silence. Following the terminology in Montello et al. (2025), we refer to neural networks which *can* vary compute based on either self-estimated difficulty or external conditions as *dynamic*, while networks that cannot are *static*. One method for introducing dynamism in neural networks is *early exit* (Scardapane et al., 2020b), where the network can be used to make predictions at several depths through the network stack based on an exit condition. Other methods include *dynamic routing* where various subsets of the network weights are used during inference

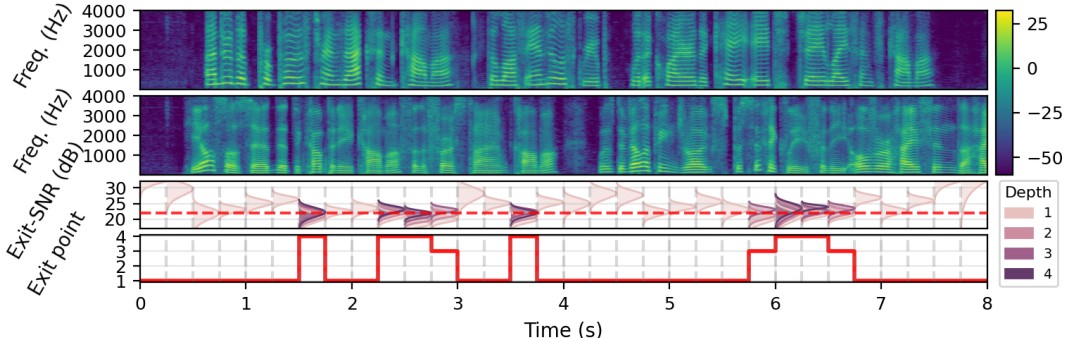

Figure 1: **Reconstructed spectrograms** of two speakers from the WSJ0-2mix test set separated by a PRESS-4-S model with 4 exit points evaluated in segments of $T = 2000$ samples, showing our proposed exit-SNR exit condition evaluated for each segment with a target level of 22 dB (shown in red). The distributions of each exit-SNR condition is shown shaded by exit point, demonstrating non-trivial improvement for deeper exits. An extended version showing our other SNR-like distributions can be seen in appendix B, fig. 7.

based on a routing condition, such as mixture-of-experts (Mu & Lin, 2025) (MoE) and slimmable networks (Yu et al., 2018).

Early-exit allows direct scaling of both compute and the total number of activated parameters in the network, yielding both energy and prediction latency savings (Jazbec et al., 2024), while allowing fine-grained decision-making using all information and processing available up to the current exit point. End-to-end training of early-exiting neural networks has also been shown to reduce overfitting and improve gradient propagation during training (Scardapane et al., 2020b). In the context of speech separation, a SepReformer (Shin et al., 2024) trained with additional reconstruction losses reminiscent of early exit improved the final performance of the network.

Prior work on early exit typically defines the exit condition implicitly through the loss function by defining a reconstruction loss and a utilization loss and minimizing a convex combination of the two (Tan et al., 2023; Bralios et al., 2023; Elminshawi et al., 2024), which leads to an implicit performance-compute trade-off that is frozen during training, and hence cannot be adapted at inference time. Other work instead defines the exit condition based on similarity between consecutive exit points (Chen et al., 2020), which does not ground the exit condition in any performance metric (which the first kind does through the loss).

**Contributions** In this work, we introduce PRobabilistic Early-exit for Speech Separation (PRESS) which leverages early exits to reduce inference costs. Importantly, our approach is probabilistic and naturally assign weights to the different exit conditions by their quantified log-likelihood-based reconstruction quality. At inference time, the probabilistic formulation enables PRESS to control when to exit computation using the model's confidence that a desired signal-to-noise ratio (SNR) between the estimated and clean audio signals has been achieved, providing a directly interpretable exit condition grounded in the network's performance at each exit point.

Specifically, we make the following contributions:

1. We propose a new early-exit framework with uncertainty awareness though a probabilistic model formulation accounting for both the clean speech signal and the variance of the error as well as its associated uncertainty. We demonstrate that our approach provides a simple and principled framework to balance optimization of reconstruction quality with early-exit accuracy without the need for careful weighting of multiple objectives. Using the probabilistic model we construct several fully probabilistic SNR-like early-exit conditions which can be used to early-exit when a target SNR level has been reached with a given uncertainty tolerance.

2. To instantiate our framework, we propose a new speech separation architecture based on linear recurrent neural networks (RNNs) designed with the joint goals of (1) achieving SOTA-level reconstruction performance per compute, and (2) being architecturally capable of outputting high-quality reconstructions from its early-exit points.

3. We validate our approach for both speech separation on the WSJ0-2mix (Garofolo et al., 2007), Libri2Mix (Tzinis et al., 2022), WHAM! (Wichern et al., 2019), WHAMR! (Maciejewski et al., 2020) datasets, and for speech enhancement on the DNS Challenge 2020 (Reddy et al., 2020) dataset, demonstrating the viability of training a single, dynamic neural network with early exits to achieve performance competitive with SOTA static, single-exit models.

## 2    RELATED WORK

**TasNet-family**    TasNet (Luo & Mesgarani, 2018) replaced masking in a fixed spectral representation with learnable convolutional encoder and decoder layers. The bulk of parameters and compute of TasNets are in a RNN masker network, which produces masks that are applied to the encoded mixture before decoding to produce single-source estimates. TasNet inspired follow-up work such as computationally efficient variants like the Conv-TasNet (Luo & Mesgarani, 2019) and SudoRMRF (Tzinis et al., 2020), as well as the DP-RNN (Luo et al., 2020) which introduced dual-path processing, decoupling time-mixing operations into chunks. The dual-path structure was also used with transformer-based networks in SepFormer (Subakan et al., 2021), TF-GridNet (Wang et al., 2023) and MossFormer (Zhao & Ma, 2023; Zhao et al., 2023).

**SepReformer**    SepReformer (Shin et al., 2024) introduces *early splitting*; instead of projecting a shared encoded representation into separate estimated sources late in the network as in TasNet, the projection into separate sources occurs early in the network. Further processing happens independently for each source with only a cross-speaker attention layer enabling information exchange between processing for the sources. SepReformer is constructed as a U-net (Ronneberger et al., 2015) with a transformer-like stack (Vaswani et al., 2023) composed of a convolutional block, a time-mixing attention block, and a cross-speaker attention block, and uses LayerScale (Touvron et al., 2021) to train comparatively deep and slim networks. The network maximizes SI-SNR using uPIT with extra loss terms maximizing the SI-SNR of downsampled frequency-domain representations of the estimated sources to encourage early separation of sources.

**Diffusion models, SNR estimation and iterative refinement**    SepIt (Lutati et al., 2023) iteratively refines its estimates and stops processing based on bounding the SNR through mutual information between current estimates and the input mixture—this bound is extended to consider generative models in DiffSep (Scheibler et al., 2022). Diffusion/score-based models like DiffSep can, generally, sample the learned diffusion process with variable compute requirements in a trade-off with quality. For instance, DiffWave (Kong et al., 2021) — a diffusion-based vocoder also used in separation tasks in Separate and Diffuse Lutati et al. (2023)—shows faster sampling by reducing needed steps through designing an appropriate variance schedule.

**Slimmable Networks**    allow the *width* of the network to be adjusted at inference time to trade-off compute and accuracy have been termed slimmable networks (Yu et al., 2018; Yu & Huang, 2019; Li et al., 2021). This can be achieved using switchable batch normalization (Yu et al., 2018), which essentially, given a predefined set of widths, trains a single network with multiple batch normalization layers, one pr. width. The width specific batch-normalization can be moved to post-training (Yu & Huang, 2019), employing knowledge distillation during training while sampling different widths pr. batch. The idea of *dynamic* slimmable networks (Li et al., 2021) uses a dynamic gating mechanism that routes the signal to a subset of the next stage in the network to reduce complexity. In the context of speech and audio, Slim-TasNet (Elminshawi et al., 2023) combines the TasNet architecture for speech separation with width slimming and a given input width size. A dynamic version of this, dynamic slimmable TasNet (Elminshawi et al., 2024), utilizes a predictive scheme, where each subnetwork predicts how much of the following subnetwork that should be utilized. Similarly, in the context of speech enhancement, dynamic channel pruning (Miccini et al., 2023) is a technique for estimating a mask applied in the channel dimension of convolutional layers, that reduces the runtime computational cost.

**Early Exit**    Various strategies are used to construct early-exit conditions, and to train early-exit models. In classification tasks, such as image classification, the entropy at each exit has been used as an exit condition (Teerapittayanon et al., 2017; Scardapane et al., 2020a) using pre-softmax logits as a proxy for model uncertainty. In the context of language modeling the Sparse Universal

Transformer (Tan et al., 2023) used a stick-breaking construction to define monotonically increasing halting probabilities which down-weighted intermediate activations in later layers to scale down their influence on the final prediction, which was learned through a single loss. In the context of speech separation, the Euclidean norm difference between successive blocks has been used as an exit condition in a transformer-based architecture (Chen et al., 2020) and early-exit has also been used based on a learned gating function in an iterative model (Bralios et al., 2023). In speech enhancement, PDRE (Nakatani et al., 2025) iteratively applies a deterministic U-net network to the noisy signal and predicts the parameters of a Gaussian mixture model (GMM), to get a distribution over the clean signal at each iteration, and the network is trained to maximize the weighted sum of log-likelihoods of each step's GMM. However, no stopping criteria or exit-conditions were explored to determine when to stop the enhancement process.

## 3 METHODS

The goal of single-channel speech separation neural networks (and enhancement as a special case) is to separate an input mixture signal $\widetilde{\boldsymbol{x}} \in \mathbb{R}^T$ into a set of estimated sources $\widehat{\boldsymbol{x}}_i \in \mathbb{R}^T$ which approximate the target sources $\boldsymbol{x}_j \in \mathbb{R}^T$ in the input mixture, where $S$ is the total number of sources and $i, j \in [1, S]$. To incorporate early-exit into such networks, we require (1) a set of early-exit *conditions* which can be used to decide when an estimate is acceptable enough to exit, (2) an *objective* to learn both the source reconstruction and the exit conditions, and (3) a neural network *architecture* which can support early exit without compromising reconstruction performance.

### 3.1 PROBABILISTIC SPEECH MODELLING

The performance of speech separation and enhancement systems is often reported as signal-to-noise ratios (SNRs) or SNR improvements (SNRis) (i.e. the relative gain in SNR by using $\widehat{\boldsymbol{x}}_i$ over the input signal $\widetilde{\boldsymbol{x}}$), computed as the ratio of the power of the target signal $\boldsymbol{x}_j$ to be estimated and the power of the error signal $\boldsymbol{x}_j - \widehat{\boldsymbol{x}}_i$,

$$\mathrm{SNR}(\boldsymbol{x}_j, \widehat{\boldsymbol{x}}_i) = \frac{\|\boldsymbol{x}_j\|_2^2}{\|\boldsymbol{x}_j - \widehat{\boldsymbol{x}}_i\|_2^2}, \qquad \mathrm{SNRi}(\boldsymbol{x}_j, \widehat{\boldsymbol{x}}_i, \widetilde{\boldsymbol{x}}) = \frac{\mathrm{SNR}(\boldsymbol{x}_j, \widehat{\boldsymbol{x}}_i)}{\mathrm{SNR}(\boldsymbol{x}_j, \widetilde{\boldsymbol{x}})} = \frac{\|\boldsymbol{x}_j - \widetilde{\boldsymbol{x}}\|_2^2}{\|\boldsymbol{x}_j - \widehat{\boldsymbol{x}}_i\|_2^2}. \qquad (1)$$

Using SNR and SNRi as early-exit conditions for speech separation is desirable because they conceptually measure the loudness of the error relative to the loudness of the target (or the improvement over the target), ensuring that the system can optimistically exit when an acceptable speech-to-noise balance is obtained.

Both SNR and SNRi however require access to the target $\boldsymbol{x}_j$, which is unknown in a predictive setting. We therefore propose to probabilistically model both the target $\boldsymbol{x}_j$ and the error of the prediction using a simple Bayesian objective where the target signal $\boldsymbol{x}_j$ is modelled by jointly predicting an estimated signal $\widehat{\boldsymbol{x}}_i$ and a variance parameter $\sigma_i^2 \in \mathbb{R}$, where we assume a Gaussian distribution on the signal error and a conjugate inverse-gamma prior on the variance. Marginalizing out the variance, we obtain a multivariate Student t-likelihood:

$$\mathcal{L}_i = \int \mathcal{N}\big(\boldsymbol{x}_j \mid \widehat{\boldsymbol{x}}_i, \sigma_i^2 \mathbf{I}\big) \mathrm{InvGam}\big(\sigma_i^2 \mid \alpha_i, \beta_i\big) \, \mathrm{d}\sigma_i^2 = \mathrm{St}\left( \boldsymbol{x}_j \, \Big| \, \widehat{\boldsymbol{x}}_i, 2\alpha_i, \frac{\beta_i}{\alpha_i} \mathbf{I} \right), \qquad (2)$$

$$\ln \mathcal{L}_i \propto \ln \Gamma\left( \alpha_i + \frac{T}{2} \right) - \ln \Gamma(\alpha_i) - \frac{T}{2} \ln \beta_i - \left( \alpha_i + \frac{T}{2} \right) \ln \left( 1 + \frac{\|\boldsymbol{x}_j - \widehat{\boldsymbol{x}}_i\|_2^2}{2\beta_i} \right), \qquad (3)$$

where $\Gamma(\cdot)$ is the gamma function and $\alpha_i$ and $\beta_i$ are the shape and scale in the inverse-gamma parameterization, which are to be predicted by the model along with the estimated signal $\widehat{\boldsymbol{x}}_i$. This objective strikes a simple balance between reducing the ratio of the signal error and variance scale through the last term, while also being penalized for underestimating the variance by the second-to-last term. See appendix F for a scale-invariant version of this objective and relationship to SI-SNR. When optimizing this objective we use uPIT to assign targets $\boldsymbol{x}_j$ to estimated sources and parameters $\widehat{\boldsymbol{x}}_i, \alpha_i, \beta_i$ by taking the maximum likelihood permutation. When training with multiple exits, we jointly permute all exits together such that speakers cannot swap between consecutive exits, and optimize the total likelihood by summing over all exits and speakers without any weighting.

**Predictive Signal-to-Noise Ratios**  Using the distributional assumptions on the target and error signals allows us to construct early exit conditions expressed directly as predictive signal-to-noise ratios, providing an interpretable thresholding mechanism for early exit. Based on our model assumptions, we have

$$\boldsymbol{x}_j \sim \mathcal{N}\big(\widehat{\boldsymbol{x}}_i, \sigma_i^2 \mathbf{I}\big), \qquad \|\boldsymbol{x}\|_2^2 \sim \sigma_i^2 \chi_T^2\left(\frac{\|\widehat{\boldsymbol{x}}_i\|_2^2}{\sigma_i^2}\right), \tag{4}$$

$$\|\boldsymbol{x}_j - \widehat{\boldsymbol{x}}_i\|_2^2 \sim \sigma_i^2 \chi_T^2, \qquad \|\boldsymbol{x}_j - \widetilde{\boldsymbol{x}}\|_2^2 \sim \sigma_i^2 \chi_T^2\left(\frac{\|\widehat{\boldsymbol{x}}_i - \widetilde{\boldsymbol{x}}_i\|_2^2}{\sigma_i^2}\right), \tag{5}$$

where $\chi_T^2(\lambda)$ is a non-central chi-square distribution with non-centrality parameter $\lambda$ and $T$ degrees of freedom. The SNR and SNRi can now be written as ratios of (non-central) chi-square distributions,

$$\mathrm{SNR}(\boldsymbol{x}_j, \widehat{\boldsymbol{x}}_i) = \frac{\|\boldsymbol{x}_j\|_2^2}{\|\boldsymbol{x}_j - \widehat{\boldsymbol{x}}_i\|_2^2} = \frac{\phi_{\mathrm{SNR}}}{\epsilon}, \qquad \phi_{\mathrm{SNR}} \sim \chi_T^2\left(\frac{\|\widehat{\boldsymbol{x}}_i\|_2^2}{\sigma_i^2}\right), \qquad \epsilon \sim \chi_T^2, \tag{6}$$

$$\mathrm{SNRi}(\boldsymbol{x}_j, \widehat{\boldsymbol{x}}_i, \widetilde{\boldsymbol{x}}) = \frac{\|\boldsymbol{x}_j - \widetilde{\boldsymbol{x}}_i\|_2^2}{\|\boldsymbol{x}_j - \widehat{\boldsymbol{x}}_i\|_2^2} = \frac{\phi_{\mathrm{SNRi}}}{\epsilon}, \qquad \phi_{\mathrm{SNRi}} \sim \chi_T^2\left(\frac{\|\widehat{\boldsymbol{x}}_i - \widetilde{\boldsymbol{x}}\|_2^2}{\sigma_i^2}\right). \tag{7}$$

These chi-square distributions are not independent, but the ratio of even dependent chi-square variables with equal degrees of freedom quickly concentrates around its mean for large $T$ (see appendix A for details), and so in the limit of large $T$ we approximate the ratios with their conditional means and see that the expressions takes the form of shifted gamma distributions,

$$\mathrm{SNR}(\boldsymbol{x}_j, \widehat{\boldsymbol{x}}_i) \xrightarrow{T \to \infty} 1 + \frac{\|\widehat{\boldsymbol{x}}_i\|_2^2}{T\sigma_i^2} = 1 + z_{\mathrm{SNR}}, \qquad z_{\mathrm{SNR}} \sim \mathrm{Gam}\left(\alpha_i, \frac{\|\widehat{\boldsymbol{x}}_i\|_2^2}{\beta_i T}\right), \tag{8}$$

$$\mathrm{SNRi}(\boldsymbol{x}_j, \widehat{\boldsymbol{x}}_i, \widetilde{\boldsymbol{x}}) \xrightarrow{T \to \infty} 1 + \frac{\|\widehat{\boldsymbol{x}}_i - \widetilde{\boldsymbol{x}}\|_2^2}{T\sigma_i^2} = 1 + z_{\mathrm{SNRi}}, \qquad z_{\mathrm{SNRi}} \sim \mathrm{Gam}\left(\alpha_i, \frac{\|\widehat{\boldsymbol{x}}_i - \widetilde{\boldsymbol{x}}\|_2^2}{\beta_i T}\right). \tag{9}$$

**Unified Early-Exit SNR**  Both SNR and SNRi can be overly pessimistic as exit conditions when used in isolation, as silence in the target such that $\|\boldsymbol{x}_j\|_2^2 \approx 0$ will cause the local SNR to approach zero, and a lack of interfering sources such that $\|\boldsymbol{x}_j - \widetilde{\boldsymbol{x}}\|_2^2 \approx 0$ similarly causes the SNRi to vanish. Additionally, in the case of total silence in all sources both SNR and SNRi vanish. To mitigate these silence issues, we propose to use a third auxiliary loudness condition measuring the SNR between a fixed reference signal $\boldsymbol{x}_{\mathrm{ref}} \in \mathbb{R}^T$ with average power $P_{\mathrm{ref}}^2 \in \mathbb{R}$ and the predicted noise signal,

$$\mathrm{SNR}_{\mathrm{ref}}(\boldsymbol{x}_j) = \frac{\|\boldsymbol{x}_{\mathrm{ref}}\|_2^2}{\|\boldsymbol{x}_j - \widehat{\boldsymbol{x}}_i\|_2^2} \xrightarrow{T \to \infty} \frac{P_{\mathrm{ref}}^2}{\sigma_i^2} = z, \qquad z \sim \mathrm{Gam}\left(\alpha_i, \frac{P_{\mathrm{ref}}^2}{\beta_i}\right). \tag{10}$$

This third condition can be used to exit whenever the predicted noise is quieter than the reference signal by some level difference, where $P_{\mathrm{ref}}^2$ should be set based on some tolerable lower bound on the loudness of the noise.

We can now integrate all three exit conditions into a single unified condition, where we consider the maximum of the individual complementary CDFs for a target SNR level $t \in \mathbb{R}$,

$$p(\mathrm{SNR}_{\mathrm{exit}}(\boldsymbol{x}_j, \widehat{\boldsymbol{x}}_i, \widetilde{\boldsymbol{x}}) \geq t) = \max \left\{ \begin{array}{l} p(\mathrm{SNR}(\boldsymbol{x}_j, \widehat{\boldsymbol{x}}_i) \geq t), \\ p(\mathrm{SNRi}(\boldsymbol{x}_j, \widehat{\boldsymbol{x}}_i, \widetilde{\boldsymbol{x}}) \geq t), \\ p(\mathrm{SNR}_{\mathrm{ref}}(\boldsymbol{x}_j) \geq t) \end{array} \right\}, \tag{11}$$

which corresponds to *optimistically* exiting when at least one of the conditions is met with sufficient confidence. We note that $p(\mathrm{SNR}_{\mathrm{exit}} \geq t)$ is itself a valid complementary CDF whose full distribution can be given in terms of its individual component PDFs and CDFs.

To obtain our final combined exit condition across all speakers we take the minimum over the complementary exit-SNR CDFs for all speakers and exit when the target level $t$ is exceeded with confidence $p \in [0, 1]$,

$$\min_i p(\mathrm{SNR}_{\mathrm{exit}}(\boldsymbol{x}_j, \widehat{\boldsymbol{x}}_i, \widetilde{\boldsymbol{x}}) \geq t) \geq p, \tag{12}$$

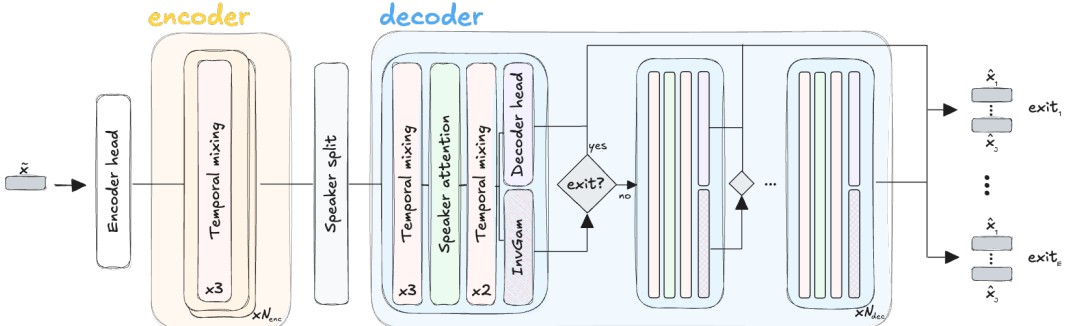

Figure 2: **Detailed architecture** of PRESS-Net. It consists of three parts: an encoder, an early split module and a reconstruction decoder with the ability to reconstruct early.

which corresponds to *pessimistically* exiting only when all speaker meet at least one condition with sufficient confidence. We show the predicted distributions of this condition in fig. 1, where $P_{\text{ref}}^2$ is set to $-35$ decibel (dB) relative to full scale (dBFS) such that a target level of $t = 22$ dB corresponds to an estimated noise loudness below $-57$ dBFS, and a full version in appendix B, fig. 7 where the individual component distributions are also shown. We leave the extension of early-exiting each speaker individually as important future work.

**Calibration** Using to our probabilistic modelling setup we can readily investigate whether our predicted $\sigma^2$ distributions are calibrated — that is, they model the actual distribution of the observed mean error power $\widehat{\sigma}_i^2 = \frac{\|\boldsymbol{x}_j - \widehat{\boldsymbol{x}}_i\|_2^2}{T}$ well. We consider two methods for evaluating this: (1) the probability integral transform (PIT) (Dheur & Taieb, 2023) evaluates the predicted CDF on the observed mean error $\widehat{\sigma}_i^2$ i.e. computing $Z_i = F_{\sigma_i^2}(\widehat{\sigma}_i^2)$, which, given perfect calibration, should itself be uniformly distributed — we use this property to construct calibration curves in fig. 5, and (2) the continuous ranked probability score (CRPS) (Matheson & Winkler, 1976) which is a proper scoring rule generalizing the mean absolute error to distributional regression.

**Block-wise Likelihood** Modelling a single, global $\sigma_i^2$ assumes a stationary error signal which is not realistic for real-world audio. We also experiment with a blocked variant of the likelihood in eq. (2), where we model blocks of $T$ consecutive samples by independent $\sigma_{i,b}^2$ for every block $b$. This redefines $T$ to a fixed block size rather than the full length of the audio signal. Note that this likelihood does not require the underlying neural network to be evaluated in blocks, and only serves to control which samples in the estimated sources are modelled by the corresponding predicted block-wise inverse-gamma parameters.

## 3.2 MODEL ARCHITECTURE

We design a TasNet-family model suitable for early exit using PRESS, and denote the architecture PRESS-Net as illustrated in fig. 2. We base the initial design of the PRESS-Net architecture on building blocks from the SepReformer (Shin et al., 2024) model due to its strong performance in speech separation. Following recent trends (Shin et al., 2024; Wang et al., 2023), we opt for an encoder-separator-decoder design where a shallow encoder/decoder pair (referred to as encoder/decoder heads in our models) down-/upsamples the audio signals whereas a separator module directly maps the encoder output to the decoder, rather than masking the encoder output as in TasNet. The encoder/decoder heads and inverse-gamma parametrization blocks are described in appendix C. We do not perform any down- or upsampling in the separator network in order to be able to reconstruct estimated sources early in the network without excessive artifacts from upsampling. We use the GELU (Hendrycks & Gimpel, 2023) activation function.

**Separator** We construct our separator module as a deep transformer-like stack with pre-norm, skip connections, and LayerScale (Touvron et al., 2021; Wortsman et al., 2023). Specifically, for every layer $f$ with input $\boldsymbol{x}$ in the stack we compute its output as $\boldsymbol{x} \leftarrow \boldsymbol{x} + \boldsymbol{\gamma} f(\text{norm}(\boldsymbol{x}))$, where $\text{norm}(\cdot)$ is an RMSNorm (Zhang & Sennrich, 2019) layer with per-channel scales initialized to 1

| Ablation | SI-SNRi | SDRi | # Params |
|---|---|---|---|
| (a)  SI-SNR loss | 22.95 | 23.1 | 3.55M |
| (b)  Normal likelihood loss | 22.42 | 22.58 | 3.55M |
| (c)  t-likelihood + per-exit uPIT | 21.1 | 20.97 | 3.55M |
| (d)  t-likelihood + 6 exits | 22.89 | 23.01 | 3.57M |
| (e)  t-likelihood + 12 exits | 22.9 | 22.99 | 3.66M |
| (f)  t-likelihood + 200K finetune | 22.9 | 23.11 | 3.55M |

(a) Training setup ablations.

| Ablation | SI-SNRi | SDRi | Receptive field |
|---|---|---|---|
| $T = 8000$ | 22.82 | 22.98 | 1000ms |
| $T = 4000$ | 22.81 | 22.99 | 500ms |
| $T = 2000$ | 22.79 | 22.96 | 250ms |
| $T = 1000$ | 22.69 | 22.91 | 125ms |
| $T = 500$ | 22.69 | 22.86 | 62ms |

(b) Block size ablations.

Table 1: **Ablation results** on the WSJ0-2mix test set.

and $\epsilon = 10^{-2}$ as in Parker et al. (2024), and $\gamma \in \mathbb{R}^D$ is the per-channel LayerScale scaling factors, which we initialize to $10^{-5}$. This allows very deep networks to be trained stably as the separator is approximately a skip connection at initialization and only later learns to integrate non-linearity as $\gamma$ grows.

Due to not downsampling in the separator module the time-resolution of the intermediate activations is much higher than in SepReformer, precluding the direct application of self-attention over time due to its prohibitively high cost from the quadratic compute scaling. Instead, we use linear RNNs with self-gating as our primary building block, as well as speaker attention layers from SepReformer throughout the separator. These blocks are described in more detail in appendix C, and illustrated in fig. 8.

We use *early split* as in SepReformer, where the first $N_{\text{Enc}}$ layers (consisting only of linear RNN blocks) in the stack process the encoded speech mixture after which a *speaker split* module $\text{SpeakerSplit} : \mathbb{R}^{T \times D} \to \mathbb{R}^{T \times S \times D}$ projects the refined speech mixture into $S$ separate groups of channels, where the speaker dimension is considered a batch dimension in later processing.

After splitting, we repeat a decoder block $N_{\text{Dec}}$ times which consists of linear RNN blocks and speaker attention blocks repeated in a 5:1 ratio. After each block, we may place an exit point $E_i < N_{\text{Dec}}$ which reconstructs the current latent representation with a separate decoder head. Apart from reconstruction, each exit point parametrizes the inverse gamma distribution which we can use to evaluate whether we should exit or not.

## 4   RESULTS AND DISCUSSION

PRESS is evaluated on four speech separation corpora: WSJ0-2mix (Garofolo et al., 2007), Libri2Mix (Cosentino et al., 2020), WHAM! (Wichern et al., 2019), and WHAMR! (Maciejewski et al., 2020), as well as one denoising corpora: DNS2020 (Reddy et al., 2020). Complete descriptions of the corpora can be found in appendix D.

We train two main model configurations: (1) PRESS-4 (S), a smaller model with model dimension $D = 64$, $P = 4$, $D_{\text{Enc}} = 256$, $N_{\text{Enc}} = 8$, $N_{\text{Dec}} = 12$ and 4 exit points placed at every 3 decoder blocks, and (2) PRESS-12 (M), a larger model with model dimension increased to $D = 128$, $N_{\text{Enc}} = 4$, $N_{\text{Dec}} = 24$ and having 12 exits, placed at every second decoder block. Further details of model training can be found in appendix E.

We show the performance of our models in table 2 compared with other SOTA methods, and in fig. 3 we plot performance in SI-SNR improvement (SI-SNRi) as a function of giga-multiply-accumulates (GMAC) per second (GMAC/s), showing how PRESS models can scale their compute dynamically while remaining competitive on WSJ0-2Mix.

In fig. 5 we plot calibration curves for the WSJ0-2Mix training and test sets, showing that our models are not well-calibrated after training on 4-second audio clips, and experiment with finetuning these with just 200K extra training steps (ca. 3% of base training time) on full-length audio clips from the training set, after which we see our models become well-calibrated, and performance also increases substantially which we show in table 2 in the bottom rows.

We train several ablation variants of PRESS-4 (S) to determine the effectiveness of (a) using SI-SNR as loss instead of our t-likelihood, (b) using a normal likelihood with a single predicted variance

| Model | WSJ0-2mix | | Libri2Mix | | WHAM! | | WHAMR! | | # Params | GMAC/s |
|---|---|---|---|---|---|---|---|---|---|---|
| | SI-SNRi (dB) | SDRi (dB) | SI-SNRi (dB) | SDRi (dB) | SI-SNRi (dB) | SDRi (dB) | SI-SNRi (dB) | SDRi (dB) | (M) | (G/s) |
| Conv-TasNet[†] | 15.3 | 15.6 | 12.2 | 12.7 | 12.7 | – | 8.3 | – | 5.1 | 10.5 |
| DualPathRNN | 18.8 | 19.0 | 16.1 | 16.6 | 13.7 | 14.1 | 10.3 | – | 2.6 | 42.5 |
| SepFormer | 20.4 | 20.5 | 19.2 | 19.4 | 14.7 | 16.8 | 14.0 | – | 26.0 | 86.9 |
| SepFormer + DM[†] | 22.3 | 22.5 | – | – | 16.4 | 16.7 | 14.0 | 13 | 26.0 | 86.9 |
| MossFormer (S) | 20.9 | – | – | – | – | – | – | – | 10.8 | –[2] |
| MossFormer (M) + DM | 22.5 | – | – | – | – | – | – | – | 25.3 | –[2] |
| MossFormer (L) + DM | 22.8 | – | – | – | 17.3 | – | 16.3 | – | 42.1 | 70.4 |
| MossFormer2 + DM | 24.1 | – | 21.7 | – | 18.1 | – | 17.0 | – | 55.7 | 84.2 |
| TF-GridNet (S) | 20.6 | – | – | – | – | – | – | – | 8.2 | 19.2 |
| TF-GridNet (M) | 22.2 | – | – | – | – | – | – | – | 8.4 | 36.2 |
| TF-GridNet (L) | 23.4 | 23.5 | – | – | – | – | 17.3 | 15.8 | 14.4 | 231.1 |
| SepMamba (S) + DM | 21.2 | 21.4 | – | – | – | – | – | – | 7.2 | 12.5 |
| SepMamba (M) + DM | 22.7 | 22.9 | – | – | – | – | – | – | 22.0 | 37.0 |
| SepReformer (T) | 22.4 | 22.6 | 19.7 | 20.2 | 17.2 | 17.5 | – | – | 3.7 | 10.4 |
| SepReformer (S) | 23.0 | 23.1 | 20.6 | 21.0 | 17.3 | 17.7 | – | – | 4.5 | 21.3 |
| SepReformer (M) | 24.2 | 24.4 | 22.0 | 22.2 | 17.8 | 18.1 | – | – | 17.3 | 81.3 |
| SepReformer (L) + DM | 25.1 | 25.2 | – | – | 18.4 | 18.7 | 17.2 | 16.0 | 55.3 | 155.5 |
| PRESS-4 @ 4 (S) | 22.91 | 23.08 | 20.04 | 20.41 | 16.49 | 16.91 | 14.54 | 13.37 | 3.4 | 11.3 |
| PRESS-12 @ 4 (M) | 22.64 | 22.93 | 19.75 | 19.71 | 16.43 | 16.71 | 14.24 | 13.09 | 8.7 | 29.1 |
| PRESS-12 @ 8 (M) | 23.47 | 24 | 20.42 | 20.86 | 16.57 | 17.03 | 14.67 | 13.45 | 15.6 | 54.4 |
| PRESS-12 @ 12 (M) | 24.28 | 24.46 | 20.88 | 21.31 | 16.65 | 17.12 | 14.69 | 13.47 | 22.4 | 79.7 |
| PRESS-4 @ 4 (S) + FT | 23.41 | 23.56 | 21.01 | 21.36 | 17.25 | 17.58 | 15.13 | 13.92 | 3.4 | 11.3 |
| PRESS-12 @ 4 (M) + FT | 23.27 | 23.43 | 20.31 | 20.72 | 16.47 | 16.81 | 14.91 | 13.73 | 8.7 | 29.1 |
| PRESS-12 @ 8 (M) + FT | 24.18 | 24.40 | 20.92 | 21.33 | 17.21 | 17.34 | 15.61 | 14.38 | 15.6 | 54.4 |
| PRESS-12 @ 12 (M) + FT | 24.36 | 24.55 | 21.29 | 21.68 | 17.49 | 17.89 | 15.67 | 14.43 | 22.4 | 79.7 |

Table 2: **Speech separation performance** on the WSJ0-2Mix, Libri2Mix, WHAM! and WHAMR! 2-speaker test sets with our PRESS variants highlighted with a gray background. We evaluate our PRESS-4 (S) model on its final exit point, and also show results for our larger PRESS-12 (M) model at 3 exit points. +DM: models trained with dynamic mixing. +FT: models finetuned on full-length training data. [†]: some values from Shin et al. (2024).

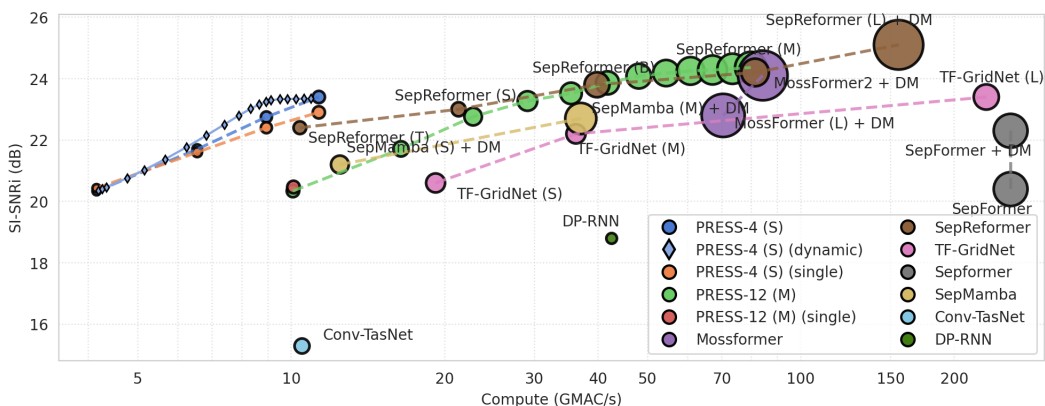

Figure 3: **Source separation performance** on WSJ0-2mix in terms of SI-SNR improvement (SI-SNRi) per compute (giga-multiply-accumulates (GMAC) per second (GMAC/s)), with the area of points corresponding to parameter count of models. The static performance of every exit point is shown for PRESS models, as well as the dynamic performance of the PRESS-4 (S) model using our probabilistic exit condition for varying target levels, beating the static performance curve in efficiency. We also include the performance of single-exit models, which underperform the jointly trained model at deeper exits.

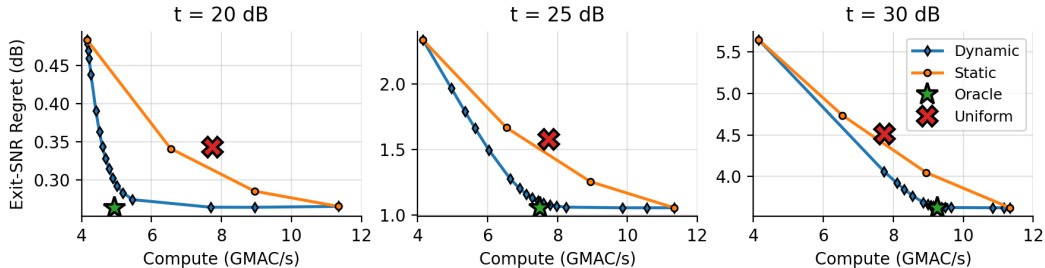

Figure 4: **One-sided exit-SNR regret** on the WSJ0-2mix test set for a PRESS-4 (S) model trained with a block size of 2000 samples using different early-exit strategies with target levels of $t = 20, 25, 30$ dB: **(dynamic)** our probabilistic exit strategy in eq. (11) evaluated for varying confidence thresholds $p$, **(static)** using a single exit for all blocks, **(oracle)** a best-case strategy that always exits when the target is achieved using the ground-truth exit-SNR, **(uniform)** an uninformed strategy that selects exit points uniformly at random.

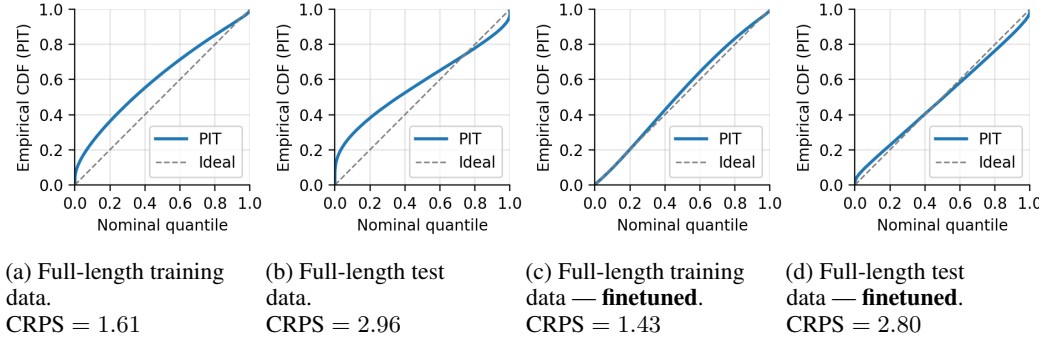

(a) Full-length training data.
CRPS $= 1.61$

(b) Full-length test data.
CRPS $= 2.96$

(c) Full-length training data — **finetuned**.
CRPS $= 1.43$

(d) Full-length test data — **finetuned**.
CRPS $= 2.80$

Figure 5: **Calibration curves** for the predicted $\sigma_i^2$ mean error distributions on the WSJ0-2mix test set for a PRESS-4 (S) model with a block size of 2000 samples. In **(a)** and **(b)** we see that the distributions are uncalibrated when the model is trained on 4-second clips and evaluated on full-length sequences on both training and test data. In **(c)** and **(d)** we see that the model predictions become well-calibrated on both training and test data after finetuning on full-length training data.

instead of a normal-inverse-gamma likelihood (c) joint (i.e. all per-exit losses permuted together) vs. per-exit permutation-invariance where speakers can swap between exits, (d) and (e) the number of exits — we train 4-, 6-, and 12-exit variants of our small model configuration with exit-points placed uniformly over depth, and (f) where we ablate that simply finetuning on more 4-second audio does not lead to the same performance improvements as training on full-length data.

Ablation results can be found in table 1. For ablation (a) we see that our Student t-likelihood can be used in place of SI-SNR without loss of performance, even though our t-likelihood is not scale-invariant. Ablation (b) shows that a simpler normal likelihood results in worse reconstruction performance in terms of SI-SNR, possibly due to not log-scaling the error as both SI-SNR and the t-likelihood do. Ablation (c) reveals that permuting consecutive early exits together is crucial for stable joint training of early exits, likely because per-exit permutation-invariance would allow the network to swap sources through the speaker attention layers, defeating much of the point of the early split architecture design. In ablations (d) and (e) we found that increasing the number of exits from 4 to 6 or 12 did not worsen performance at any of the exit points, which motivated us to train larger models at the 12-exit level. We also see in ablation (f) that additional finetuning with 4-second audio does not increase performance.

We further investigate the use of the PRESS architecture the DNS2020 dataset by treating the speech enhancement task as a source separation, where we predict both the clean speech and noise signals as separate sources. Surprisingly, as seen in table 3, this leads to very competitive performance after accounting for total GMAC/s even though our model also explicitly recovers the noise signal

|  | SI-SDR | STOI | WB-PESQ | # Params | GMAC/s |
|---|---|---|---|---|---|
| MFNet (Liu et al., 2023) | 20.31 | 97.98 | 3.43 | – | 6.1 |
| MP-SENet (Lu et al., 2023) | 21.03 | 98.16 | 3.62 | 2.26M | 40.7 |
| TF-Locoformer (Saijo et al., 2024) | 23.30 | 98.80 | 3.72 | 14.97M | 248.6 |
| ZipEnhancer (Wang & Tian, 2025) | 22.22 | 98.65 | 3.81 | 11.34M | 133.5 |
| PRESS-4 @ 4 (S) | 20.53 | 96.34 | 2.69 | 3.55M | 11.6 |
| PRESS-12 @ 4 (M) | 20.97 | 96.52 | 2.92 | 8.57M | 29.1 |
| PRESS-12 @ 8 (M) | 21.98 | 96.97 | 3.10 | 14.95M | 53.7 |
| PRESS-12 @ 12 (M) | 22.15 | 97.13 | 3.10 | 18.14M | 78.3 |

Table 3: **Speech enhancement performance** on the DNS2020 non-blind test set without reverberation.

while other methods do not, with our smaller PRESS-12 (M) model configuration matching the ZipEnhancer (Wang & Tian, 2025) method in terms of SI-SNRi using substantially less compute.

We also experiment with evaluating our exit-SNR condition in fig. 4, at 3 different target levels by varying the confidence threshold $p$, and measuring the one-sided regret of our exit condition, i.e. the difference between the achieved exit-SNR and the target level, or 0 if the exit-SNR exceeded the target. We see that using our early exit strategy closely matches the oracle strategy at the appropriate confidence level.

## 5 CONCLUSION & FUTURE WORK

We introduced the PRESS method and PRESS-Net model, achieving competitive performance on WSJ0-2Mix, Libri2Mix, WHAM!, WHAMR!, and DNS Challenge 2020 while allowing flexible compute scaling based on probabilistic early-exit conditions.

Our probabilistic approach allows well-defined early exit conditions to be formulated with integrated uncertainty quantification using the CDF of the exit-SNR distributions, and can be used in place of conventional training objectives such as SI-SNR at no apparent cost to the reconstruction objective.

Our predictive SNR-like distributions proved to be very well-calibrated after finetuning on full-length data, which also provided significant performance improvements to reconstruction.

An extension of our work would be to consider iterative models, i.e. special cases of our model with a single shared block in the decoder stack repeated for each exit point. This would allow theoretically infinite scaling with compute, but if done naively couples the total parameter count to the size of the iterative block requiring more careful network design, possibly using width-scaling neural networks.

## ACKNOWLEDGEMENTS

This work is partly funded by WS Audiology and the Innovation Fund Denmark (IFD) under File No. 3129-00075B.

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

# APPENDIX

## A $\chi^2$-RATIO APPROXIMATION

In section 3.1 we claim that our model assumptions allow us to express a probabilistic SNRi in terms of the estimated error variance $\sigma^2$. From our likelihood we have (omitting speaker indices $i$ and $j$),

$$\boldsymbol{x} \sim \mathcal{N}(\widehat{\boldsymbol{x}}, \sigma^2 \mathbf{I}), \qquad \sigma^2 \sim \mathrm{InvGam}(\alpha, \beta). \tag{13}$$

which we can equivalently write as,

$$\boldsymbol{x} = \widehat{\boldsymbol{x}} + \sigma \boldsymbol{z}, \qquad \boldsymbol{z} \sim \mathcal{N}(\mathbf{0}, \mathbf{I}). \tag{14}$$

We then obtain

$$\phi = \|\boldsymbol{x} - \widetilde{\boldsymbol{x}}\|_2^2 = \|\widehat{\boldsymbol{x}} + \sigma \boldsymbol{z} - \widetilde{\boldsymbol{x}}\|_2^2 = \sigma^2 \left\| \boldsymbol{z} + \frac{\widehat{\boldsymbol{x}} - \widetilde{\boldsymbol{x}}}{\sigma} \right\|_2^2 \sim \sigma^2 \chi_T^2 \left( \frac{\|\widehat{\boldsymbol{x}} - \widetilde{\boldsymbol{x}}\|_2^2}{\sigma^2} \right), \tag{15}$$

$$\epsilon = \|\boldsymbol{x} - \widehat{\boldsymbol{x}}\|_2^2 = \sigma^2 \|\boldsymbol{z}\|_2^2 \sim \sigma^2 \chi_T^2, \tag{16}$$

$$\mathrm{SNRi} = \frac{\phi}{\epsilon}. \tag{17}$$

These distributions are dependent on the same draw of $\boldsymbol{z}$ and consequently the distribution of $\frac{\phi}{\epsilon}$ does *not* take the form of a non-central F-distribution, nor can we rely on existing dependence results that assume shared sub-summations in the two associated $\chi^2$ distributions (Provost & Rudiuk, 1994) or the correlation to be known (Joarder, 2009).

Notably, the $\sigma^2$ scaling factors cancel in the ratio, so that the only sources of randomness are (1) the draw of $\sigma^2$ in the non-centrality parameter of $\phi$ and (2) the draw of $\boldsymbol{z}$. We now instead consider the limiting behavior of the ratio as $T \to \infty$ to see how it may be approximated for large $T$. Since $\|\widehat{\boldsymbol{x}} - \widetilde{\boldsymbol{x}}\|_2^2$ is a constant which scales linearly with $T$, we can rewrite the non-centrality parameter of $\phi$ to $T\lambda$, where we have absorbed the constant scale into a new random variable $\lambda = \frac{\|\widehat{\boldsymbol{x}} - \widetilde{\boldsymbol{x}}\|_2^2}{T\sigma^2} \sim \mathrm{Gam}\left(\alpha, \frac{\|\widehat{\boldsymbol{x}} - \widetilde{\boldsymbol{x}}\|_2^2}{T\beta}\right)$. We then have to show that for

$$\phi = X_T \sim \chi_T^2(T\lambda), \qquad \epsilon = Y_T \sim \chi_T^2, \tag{18}$$

the ratio $\frac{X_T}{Y_T}$ converges in distribution to

$$\frac{X_T}{Y_T} \xrightarrow{d} 1 + \lambda \quad \text{as} \quad T \to \infty. \tag{19}$$

We start by noting that the normalized variables $\frac{X_T}{T}$ and $\frac{Y_T}{T}$ have decreasing variance as $T \to \infty$,

$$\mathbb{E}\left[\frac{X_T}{T} \,\middle|\, \lambda\right] = \frac{\mathbb{E}[X_T \mid \lambda]}{T} = 1 + \lambda, \qquad \mathrm{Var}\left[\frac{X_T}{T} \,\middle|\, \lambda\right] = \frac{\mathrm{Var}[X_T \mid \lambda]}{T^2} = \frac{2 + 4\lambda}{T} = \mathcal{O}\left(\frac{1}{T}\right), \tag{20}$$

$$\mathbb{E}\left[\frac{Y_T}{T}\right] = \frac{\mathbb{E}[Y_T]}{T} = 1, \qquad \mathrm{Var}\left[\frac{Y_T}{T}\right] = \frac{\mathrm{Var}[Y_T]}{T^2} = \frac{2}{T} = \mathcal{O}\left(\frac{1}{T}\right), \tag{21}$$

which implies that both normalized variables converge in probability to their expected values as $T \to \infty$. By Slutsky's theorem (Casella & Berger, 2024), the ratio $\frac{X_T}{Y_T}$ then also converges in distribution as,

$$\frac{X_T}{Y_T} = \frac{\frac{X_T}{T}}{\frac{Y_T}{T}} \xrightarrow{d} \frac{\mathbb{E}\left[\frac{X_T}{T} \,\middle|\, \lambda\right]}{\mathbb{E}\left[\frac{Y_T}{T}\right]} = 1 + \lambda \quad \text{as} \quad T \to \infty. \tag{22}$$

This convergence holds even when $X_T$ and $Y_T$ are not independent, as the normalized variables converge to the expected values separately. The SNR case is analogous, substituting $\|\widehat{\boldsymbol{x}} - \widetilde{\boldsymbol{x}}\|$ for $\|\widehat{\boldsymbol{x}}\|$.

To get a feeling for the rate of convergence, we simulate the true distribution of $\frac{X_T}{Y_T}$ for finite $T$ with $\lambda \sim \mathrm{Gam}\left(\alpha, \frac{c}{\beta}\right)$ and perform one-sample Kolmogorov-Smirnov tests comparing the empirical

distribution of the ratio for $1\,000\,000$ samples to the limiting distribution $1 + \lambda$ in fig. 6. We set the parameters $\alpha = 60$, $\beta = 0.1$, and $c = 0.1$ based on the average model predictions seen during training.

In table 4 we show the average Kolmogorov-Smirnov (KS) distance between approximate and simulated SNRi distributions on the WSJ0-2mix test set for our blocked-likelihood PRESS-4 (S) models, demonstrating that in practice even relatively small values of $T$ are still approximated well by our asymptotic distribution.

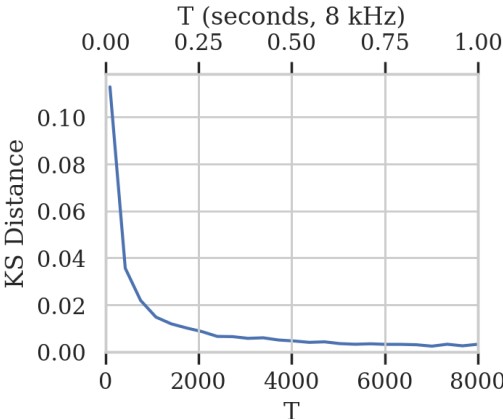

Figure 6: **Simulated KS distances** between the simulated and approximate SNRi distributions for varying values of $T$. The KS statistic represents the maximum deviation between the empirical and limiting CDFs, which is below 1% after $T \approx 2000$.

| Ablation | KS distance | Receptive field |
|---|---|---|
| Block size $T = 8000$ | 0.007 | 1000ms |
| Block size $T = 4000$ | 0.008 | 500ms |
| Block size $T = 2000$ | 0.012 | 250ms |
| Block size $T = 1000$ | 0.019 | 125ms |
| Block size $T = 500$ | 0.049 | 62ms |

Table 4: **Test KS distances** on the WSJ0-2mix test set.

## B   FULL EARLY EXIT DISTRIBUTIONS

A complete version of fig. 1 can be seen in fig. 7.

## C   MODEL ARCHITECTURE DETAILS

**Audio Encoder and Decoder Heads**   We base our encoder and decoder heads on the design from SepReformer, with the audio encoder head processing the time-domain audio signal $x$ as $\mathbb{R}^T \to \mathbb{R}^{D_{enc} \times T/P}$ by passing it through a 1-D convolution with kernel size $K = 16$, stride $P = 4$ and encoding dimension $D_{enc} = 256$ with bias, which is used to avoid division-by-zero in later RM-SNorm layers that would otherwise occur if an all-zero input signal was passed into the model. The representation is then passed through a GELU, RMSNorm and finally a linear layer which projects down to the model dimension $D_{model} = 64$ or $128$. See fig. 8d.

The decoder head consists of a gated linear unit (GLU) layer followed by a transposed convolution with kernel size $K = 16$, and stride $P = 4$ that maps the latent representation back to the original sampling rate as $\mathbb{R}^{D_{model} \times T/P} \to \mathbb{R}^T$. Every exit point has its own decoder head; see fig. 8e.

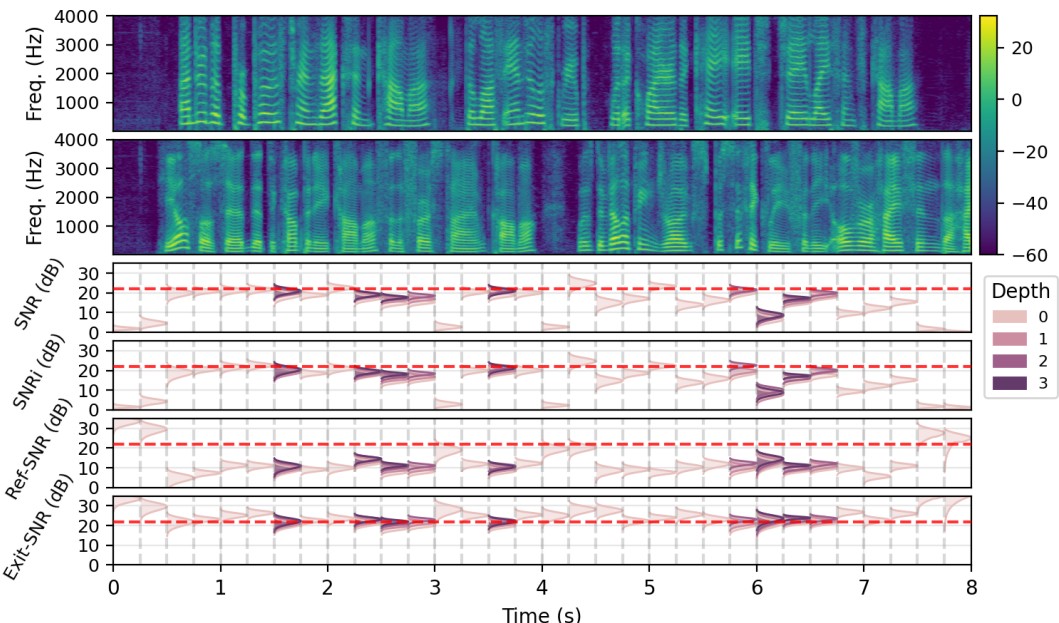

Figure 7: Full version of fig. 1 including separate exit conditions.

**Linear RNNs** To capture rich, long-range temporal relationships without the quadratic cost of attention (Vaswani et al., 2023), we use a linear RNN based on minGRU (Feng et al., 2024) and RG-LRU (De et al., 2024). The recurrence $R(\cdot, \cdot)$ takes as input $x_t \in \mathbb{R}^D$ and $r_t \in \mathbb{R}^D$ to produce an output sequence $h_t \in \mathbb{R}^D$ as follows:

$$h_t = g_t \odot h_{t-1} + (1 - g_t) \odot x_t, \qquad g_t = \sigma(\lambda)^{\sigma(r_t)}, \tag{23}$$

where $\lambda$ is a learnable diagonal matrix, and $\sigma(\cdot)$ is the sigmoid function. This parametrization ensures that $g_t$ is bounded between 0 and 1, making the recurrence stable. $\lambda$ is initialized such that $\sigma(\lambda)$ is uniformly distributed between 0.9 and 0.999 as in De et al. (2024).

Notice that $\lambda$ being a diagonal matrix, makes the computation of $g_t$ elementwise. Moreover, the recurrence itself operates entirely in an elementwise fashion. Since the recurrence is linear in terms of $h_t$ and since $g_t$ does not depend on $h_t$, the recurrence can be parallelized along the time dimension using a parallel associative scan (Blelloch, 1990; Martin & Cundy, 2018) for efficient training.

We use the quasiseparable matrix framework from Hwang et al. (2024), which we refer to as just Hydra bidirectionality, in order to construct a bidirectional variant of our recurrence:

$$\text{Hydra}_R(r_t, x_t) = \text{shift}(R(r_t, x_t)) + \text{flip}(\text{shift}(R(\text{flip}(r_t), \text{flip}(x_t)))), \tag{24}$$

where $\text{flip}(\cdot)$ reverses a sequence along the time dimension and $\text{shift}(\cdot)$ shifts the sequence one time index to the right, discarding the final time step and prepending the sequence with a zero vector $0 \in R^D$. Notice that $r_t$ and $r_t$ are shared in both terms of the bidirectionality, such that the Hydra approach does not require additional learned parameters. This bidirectionality has been shown to give better performance than simpler alternatives like additive or concatenative approaches (Hwang et al., 2024).

We use linear RNNs in a self-gating block that projects the input into $x_t$ and $r_t$ for use in eq. (23), applies the recurrence, multiplies the output with a multiplicative branch gated by a GELU activation, before finally projecting the output with another linear layer. This block closely resembles the block used in Mamba (Gu & Dao, 2023), but without convolutions which we found did not affect performance. See fig. 8b.

**Inverse Gamma Parametrization** We model the parameters $\alpha$ and $\beta$ of the inverse-gamma distribution in eq. (2) with the InvGam block shown in fig. 8f. The block is composed of a GLU layer followed by a GELU activation, which is then linearly projected to two scalars followed by a final softplus activation to the parameters to be non-negative.

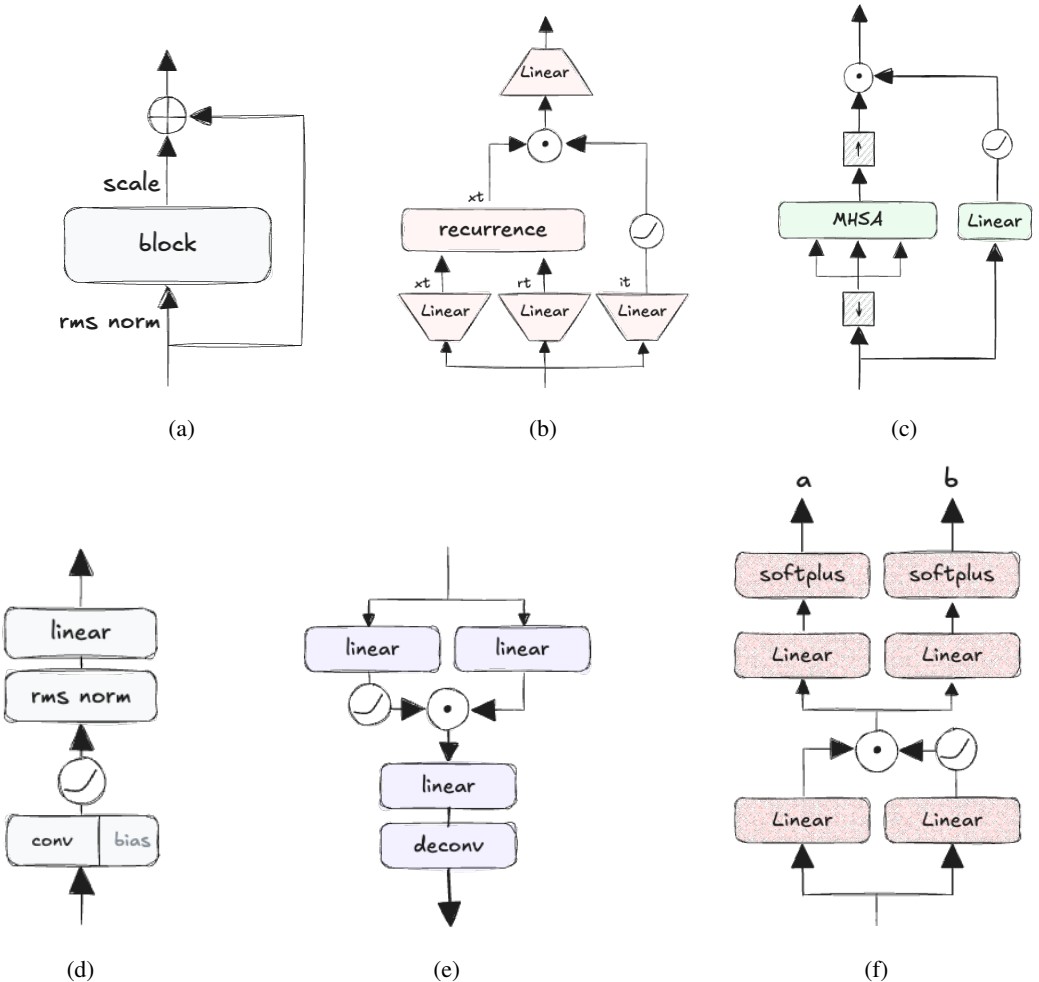

Figure 8: Detailed architecture of the blocks used. **(a)** shows the overall block architecture where the *block* which may be either **(b)** the linear RNN or **(c)** speaker attention blocks. **(d)** and **(e)** are the audio encoder and decoder heads respectively and **(f)** is the InvGamma parametrization block.

**Block-wise Early-Exits**   Early-exiting with our block-wise likelihood requires managing RNN state and reconstruction from overlapping transposed convolutions in the decoder heads. We handle RNN states by initializing all states to zero whenever an RNN layer is "restarted" after an earlier exit, and we combine contributions from different exit points across block boundaries by overlap-adding the transposed convolutions from separate layers, corresponding to smoothly interpolating between partial results from both contributions.

## D   DATASET DESCRIPTIONS

**WSJ0-2mix**   The WSJ0-2mix (Garofolo et al., 2007) dataset is generated according to a Python replica[1] of the original Matlab implementation. 2-speaker mixtures are generated by randomly selecting utterances from different speakers in the Wall Street Journal (WSJ0) corpus. The training set contains 20,000 mixtures (30 hours), the validation set has 5,000 mixtures (8 hours), and the test set comprises 3,000 mixtures (5 hours). Speech mixtures are created by mixing pairs of utterances from different speakers at random SNRs drawn uniformly between 0 and 5 dB.

**Libri2Mix**   The Libri2Mix (Cosentino et al., 2020) dataset is generated according to the official script[2], which creates 2-speaker mixtures by randomly selecting utterances from different speakers in the LibriSpeech corpus. The loudness of each individual utterance is uniformly sampled between -25 and -33 LUFS. This mixing approach results in signal-to-noise ratios (SNRs) that follow a normal distribution with a mean of 0 dB and a standard deviation of 4.1 dB. Only the train-100set, which has 40hours/9300utterances of data, is used for training and results are reported on the test set which contains 3000 samples.

**WHAM!**   The WHAM! (Wichern et al., 2019) dataset extends the WSJ0-2mix mixtures with additive environmental noise recorded in various urban environments. The mixtures are generated following the same procedure as for WSJ0-2mix, but with noise added at an SNR sampled uniformly in the -6 and 3 dB range between the loudest speaker and noise.

**WHAMR!**   The WHAMR! (Maciejewski et al., 2020) dataset in turn extends WHAM! by introducing reverberation into the WSJ0-2mix speech mixtures before adding the WHAM! noise by convolving the speech signals with artificially generated room impulse responses designed to approximate domestic and class-room environments.

**DNS2020**   The Deep Noise Suppression (DNS) Challenge 2020 (Reddy et al., 2020) dataset includes two main parts: 441 hours of clean speech extracted from LibriVox (Noa, 2005) audiobooks and a noise library of about 195 hours. The noise library combines 60,000 clips from AudioSet with 10,000 clips from Freesound (Font et al., 2013) and DEMAND (Thiemann et al., 2013) datasets. DNS2020 provides a mixing script but sets no guidelines for how much data should be mixed. Therefore, speech-noise mixtures were synthesized on the fly during training at SNRs between 0-20 dB. This synthesis followed the procedure provided in the original script[3]. Results are reported on the non-blind test set without reverberation. The test set contains 150 premixed samples.

**Sampling Rate and Downsampling**   The speech separation datasets operate at 8kHz sampling rate, while DNS2020 uses a 16kHz sampling rate. This creates the only difference between our models. For 8kHz speech separation, we use a downsampling factor of 4 in the encoder, while for the 16kHz DNS2020 we use a factor of 8, resulting in the same amount of compute per second.

## E   TRAINING DETAILS: DATASETS, RESOURCES AND SOFTWARE

We use the AdamW Kingma & Ba (2017); Loshchilov & Hutter (2019) optimizer with $\beta_1 = 0.9$ and $\beta_2 = 0.99$, and weight decay of 0.01 which we apply to only linear and convolutional layers. We use a base learning rate of $5 \cdot 10^{-4}$, which was found for a $D = 64$ model with 4 exit points,

---

[1]https://github.com/mpariente/pywsj0-mix
[2]https://github.com/JorisCos/LibriMix
[3]https://github.com/microsoft/DNS-Challenge/tree/interspeech2020/master

and we transfer this learning rate to wider models (e.g. $D = 128$) by a per-layer factor of $\frac{D_{old}}{D_{new}}$ as described in Yang et al. (2024). We use a linear straight-to-zero learning rate schedule as described in Bergsma et al. (2025) with a 5000-step linear warmup period. During training we clip the total gradient if its L2 norm exceeds 1. All weight matrices were initialized from a normal distribution truncated at $3\sigma$ with standard deviations set according to Yang et al. (2024).

For all datasets the models train on 4-second segments while at evaluation time the model process samples of varying length. The models train for up to 6 million steps with a batch size of 1, amounting to 6,666 hours of training data exposure.

We trained all models on NVIDIA GPUs with Ampere architecture or higher, using any of H100, A100, A40, A10, RTX 4090, and RTX 4070 Ti at either a university HPC cluster or commercially available GPUS with PyTorch (Paszke et al., 2019) 2.7 using `torch.compile`. The PRESS-4 (S) configurations took around 2-3 days to train, while the PRESS-12 (M) configurations took around 6 days to train.

## F    SCALE-INVARIANT STUDENT T-LIKELIHOOD

Similar to SI-SNR (Roux et al., 2018), we can construct a scale-invariant version of our likelihood by introducing a scaling parameter on the estimated signal $\widehat{x}$ which yields a modified likelihood,

$$\mathcal{L} = \mathrm{St}\left( x \,\middle|\, \gamma\widehat{x}, 2\alpha, \frac{\beta}{\alpha} \right). \tag{25}$$

Any conjugate prior can be imposed on $\gamma$, but its effect would quickly be overwhelmed by the posterior for non-trivial signal length. The maximum-likelihood estimate for $\gamma$ yields,

$$\gamma = \frac{\widehat{x}^\top x}{\widehat{x}^\top \widehat{x}}, \tag{26}$$

which closely resembles the scale-invariance parameter in the conventional SI-SNR, but normalized by the energy in the prediction rather than the target. This scale-invariance formulation has also been used with SI-SNR to train TF-GridNet (Wang et al., 2023).

Empirical studies show that TasNets trained with log-RMS error perform comparably to those trained with SI-SDR (Heitkaemper et al., 2020). While SI-SDR losses yield better models than RMS error, the key factor may be the logarithmic error scaling rather than scale invariance. Notably, the multivariate Student t-distribution likelihood in eq. (2) also measures errors on a log scale.

