# OpenReview forum: "Knowing When to Quit: Probabilistic Early Exits for Speech Separation Networks"
_ICLR.cc/2026/Conference — ICLR 2026 Poster_

### Official Review · Reviewer_fRD5 · 2025-10-21

**Soundness:** 2
**Presentation:** 3
**Contribution:** 3
**Rating:** 6
**Confidence:** 4

**Summary:**

This paper deals with the problem of single-channel speech separation, a crucial task in telecommunication, hearables, and other speech processing applications. Recent deep neural network models have achieved strong performance in this area based on commonly used metrics and loss functions such as SI-SNR and uPIT. However, neural networks with fixed computation paths may be inefficient in real-world scenarios where input signals often contain silence or non-overlapping speech. To address this, the authors propose a novel metric and loss function based on Bayesian and GMM formulations, combined with an early-exit mechanism. They extend the SepReformer architecture integrating these components to support resource-adaptive computation. Experiments on widely used benchmarks like LibriMix and WSJ0Mix demonstrate that the model achieves competitive performance comparable to SepReformer, and a favorable accuracy-efficiency trade-off with adaptive computation.

**Strengths:**

The proposed model demonstrates strong speech separation performance comparable to the state-of-the-art SepReformer, while offering support for adaptive computation and maintaining relatively low computational and memory costs.


The work provides a mathematically grounded formulation of speech separation objectives and metrics using GMM theory, offering potential alternatives to existing dominant approaches.


The evaluation includes multiple speech separation models and benchmarks, adding rigor and relevance to the empirical results.

**Weaknesses:**

The motivation and evaluation are not well aligned. The paper emphasizes potential computational savings in speech separation scenarios involving non-overlapping speech or silence; however, the evaluation only measures overall end-to-end computational efficiency. It does not quantitatively assess the specific advantages in those targeted scenarios.

The learning objective restricts the generality of the proposed framework. While metrics such as uPIT or SI-SNR can be readily applied to most speech separation architectures that output raw signals, the proposed early-exit SNR objective additionally requires the network to produce a variance output. This constraint limits applicability, as it necessitates architectural components explicitly designed for variance computation.

The reported performance improvements over existing state-of-the-art methods are marginal, and the situations where the proposed approach would provide a clear advantage are not convincingly demonstrated. In particular, the paper lacks experiments on realistic deployment settings, such as mobile or edge devices, where adaptive computation would be most beneficial. Moreover, under comparable computational budgets, the proposed model achieves slightly lower accuracy than SepReformer.

**Questions:**

For better readability among non-experts in Bayesian modeling, it would be beneficial to include a brief introduction to Bayesian Gaussian models, particularly the Normal–Inverse-Gamma (NIG) family, which appears to underpin the proposed probabilistic speech modeling framework. A concise explanation of how this family models both mean and variance uncertainty would make the paper more accessible and improve conceptual clarity.

The post-calibration procedure also seems to play a crucial role in the method’s generalizability. However, it is described as a form of training on validation data, effectively learning the variance distribution from the validation set. This raises concerns about potential data leakage or overfitting. If the model’s performance relies heavily on this post-calibration step, it would be important to clarify whether the approach generalizes well without it.

The paper introduces several architectural modifications on top of SepReformer, such as replacing the Transformer with a Linear RNN and using the Snake activation function. It would strengthen the paper to include ablation studies quantifying the individual contributions of these design choices to the overall performance.

The 2-speaker separation benchmarks on datasets such as WSJ0Mix and LibriMix are largely saturated, with SI-SNR values above 20 dB already reaching perceptual saturation. Therefore, it would be valuable to discuss whether the reported improvements, though numerically measurable, are practically significant in real-world speech separation scenarios.

---

> ### Author Response · Authors · 2025-11-25
>
> **Weakness 1: ("The motivation and evaluation are not well aligned. [...] It does not quantita-
> tively assess the specific advantages in [the] targeted scenarios.")**
>
> The goal of a strong early-exiting neural network is arguably to simultaneously identify situations where the result is already
> "good enough", and to be able to attain the desired performance.
> As such, we demonstrate both that our early-exit conditions are reliable (through calibration curves),
> and in our new results we also report standard distributional regression metrics such as continuous
> ranked probability score (CRPS) and probability integral transform (PIT) ECE, and that our method
> does not compromise reconstruction performance as is thus able to attain good performance. In our
> new results, we also include the Pareto front of our model as a function of target level in figure 3,
> where we see that actually using early-exit also does not compromise performance.
>
> **Weakness 2: ("The learning objective restricts the generality of the proposed framework")**
>
> Our method already uses uPIT to solve the permutation problem, and we do not agree that our
> method is any less general than using SI-SNR coupled with some additional mechanic to enable
> early-exit, which usually comes in the form of an auxiliary loss term that minimizes the utilization of
> the network in a reconstruction-utilization trade-off. In fact, our method is in some ways simpler: a
> single unified objective rather than an ad-hoc combination of two otherwise unrelated and potentially
> interfering objectives.
> The network components are simple prediction heads as described in appendix C, and the inverse-
> gamma predictors take up less than 10K parameters and 0.01 GMAC/s for PRESS-4-S (0.28% of
> parameters, 0.1% of compute), making them a negligible overhead.
>
> **Weakness 3: ("The reported performance improvements over existing state-of-the-art meth-
> ods are marginal")**
>
> While our method achieves raw reconstruction performance per GMAC/s
> comparable to existing SOTA, our method does this while enabling entirely new functionality at
> no cost to reconstruction, which is in itself a novel result. Using our new simplified architecture,
> our performance per GMAC/s is now consistently better than SepReformer, with PRESS-4-S nearly
> matching SepReformer (S) with only half the compute, although we stress that the main goal of our
> paper is to enable reliable early-exit.
>
> **Question 1: ("It would be beneficial to include a brief introduction to Bayesian Gaussian mod-
> els")**
>
> We agree with the reviewers suggestion, and will improve the introduction of the Bayesian
> modelling setup in our revised paper.
>
> **Question 2: ("The post-calibration procedure also seems to play a crucial role")**
>
> Based on
> our new results, the post-calibration procedure is obviated by finetuning on variable-length audio
> as described in our top level comment. Without any post-calibration and using only 200K variable-
> length training steps the PRESS-4-S model now predicts the variance distribution with an expected
> calibration error (ECE) of only 0.02 on the WSJ0-2mix test set.
> We do however not agree that our post-calibration method as described currently constitutes training
> on validation data as it fits only a single pair of scalars, making the risk of overfitting or leaking
> significant amounts of information much smaller simply due to the very low fitting capacity.
>
> **Question 3: ("The paper introduces several architectural modifications")**
>
> We have simplified
> our architecture and now use more conventional components (such as GELU in place of Snake
> activations), as described in our top level comment.
>
> **Question 4: ("WSJ0Mix and LibriMix are largely saturated")**
>
> While WSJ0-2mix performance
> is at this point largely an academic exercise (for many reasons besides saturating performance) it
> remains a useful point of common comparison to a large body of work in speech separation, and
> a similar argument holds for Libri2Mix although it is not as severely saturated yet. Performance
> improvements on WSJ0-2mix also continue to correlate well with performance on other speech
> separation datasets in the literature. We do however agree that it would strengthen the paper to also
> evaluate our methods on more realistic data, which we do in the form of reporting SI-SNRi and
> SDRi performance on WHAM! and WHAMR! in our new results.

---

### Official Review · Reviewer_9ZZr · 2025-10-29

**Soundness:** 4
**Presentation:** 4
**Contribution:** 4
**Rating:** 6
**Confidence:** 5

**Summary:**

The paper propose a neural network for single channel speech separation and enhancement.
The proposed PRESS algorithm is capable of early-exit with the ability to output uncertainty-aware probabilistic early-exit parameter.
They test it on both speech separation and enhancement tasks and shows very good results without compromising reconstruction (state of the art results).

**Strengths:**

1. The idea of early exit is novel is interesting, and the SNR based criterion is also important to the field

2. When adding more exits it save the results of the latest exit performance, and the last exit get closer results to the state of the art results.

3. The proposed method can be use in both separation task and enhancement task which is very useful.

4. The proposed method introduce the Student t-likelihood to use as likelihood measure.

**Weaknesses:**

1. The results of the proposed architecture on early exits doesn't achieves state of the art results

2. The calculation of the inverse-gamma variance at the head is global for the whole reconstruction, in real audio case one scalar is not enough to express the uncertainty

3. There may be a problem if each exit peak its own permutation since it can choose different speakers in the following exits which can lead to degradation in the training

**Questions:**

1. Can you please provide more results on speech separation tasks, i.e. on other dataset (not only WSJ-mix)?

2. Can you evaluate the proposed method on more than two speakers (3 or more)?

3. Could you consider more perceptual metric for evaluation such as STOI and PESQ?

---

> ### Author Response · Authors · 2025-11-25
>
> **Weakness 1: ("The results of the proposed architecture on early exits doesn’t achieves state of
> the art results")**
>
> Our method reaches SOTA level performance in its last exit for both PRESS-4-S
> and PRESS-4-M. With our new architecture and extended training time, we have found that PRESS-
> 4-S now much more closely matches the performance of single-exit models trained to the same size
> as each individual exit in the full early-exiting model, an ablation that we will include in the revised
> paper. We have also ablated that the reason the first exit in PRESS-4-M does not reach the same
> performance as the last exit in PRESS-4-S even though they have roughly the same GMAC/s is
> architectural — a single-exit model in PRESS-4-M configuration but with only depth up to the first
> exit achieves similar final performance to the first exit of the early-exiting model, indicating that the
> issue is likely due to a width-depth optimum in the model architecture, which we plan to add as a
> discussion point to the revised paper.
> We also highlight that even though early exits do still slightly underperform a single-exit model
> trained to the same size, this is likely a no free lunch problem where the model is overall forced to
> trade off capacity in early layers between early and later exits’ performances.
>
> **Weakness 2: ("The calculation of the inverse-gamma variance at the head is global for the
> whole reconstruction, in real audio case one scalar is not enough to express the uncertainty")**
>
> We agree with the reviewer that in real-world audio the error signal will be highly non-stationary,
> and for this reason we have experimented with a blocked version the t-likelihood as described in our
> top level comment, where the network predicts the inverse-gamma parameters for fixed-size blocks
> of the input (e.g. 2000 samples), allowing the network to appropriately model a time-varying error
> signal.
>
> **Weakness 3: ("There may be a problem if each exit [picks] its own permutation since it can
> choose different speakers in the following exits which can lead to degradation in the training")**
>
> We find exactly that per-exit permutation assignment can indeed cause instability as different ex-
> its may independently choose different source permutations at each exit point, leading to speaker
> swapping and degraded gradient signals.
> This is what we report in Table 1 as the "PRESS-S + t-likelihood + per-exit uPIT" ablation where
> using per-exit uPIT, i.e., allowing each exit to choose its own permutation, leads to performance
> degradation (SI-SNRi dropping to 21.1 dB), confirming the reviewers concern.
> In contrast, in our default PRESS setup, we explicitly perform joint permutation across consecutive
> exits based on the maximum of the summed per-exit likelihoods (as discussed in Section 3.1 and
> Section 5 in connection with ablations) which ensures that all exits are forced to share the same
> permutation assignment during training. This does not explicitly prevent sources swapping channels
> from exit to exit, but the model would be heavily penalized through the loss for doing so and in
> practice we never observe it. Indeed in our new ablations we see that our full early-exit PRESS-
> 4-S model matches or exceeds single-exit models, indicating that joint permutation resolves the
> permutation inconsistency issue. In our revised paper we will clarify the description of this ablation.

---

> ### Author Response · Authors · 2025-11-25
>
> **Question 1: ("Can you please provide more results on speech separation tasks, i.e. on other
> dataset (not only WSJ-mix)?")**
>
> In addition to the WSJ0-2mix results, Table 2 (in the original
> submission) already provides results on the well-studied Libri2Mix, and we have also trained mod-
> els on the WHAM! and WHAMR! datasets to include results in realistic noisy and reverberation
> conditions.
>
> **Question 2: ("Can you evaluate the proposed method on more than two speakers (3 or
> more)?")**
>
> The PRESS architecture and probabilistic formulation fully supports 3+ speakers; how-
> ever, we do not include 3-speaker (or more) experiments because, in the context of evaluating our
> early-exit mechanisms, such results would not provide additional insight beyond the two-speaker
> case for several reasons:
> Our probabilistic exit conditions operate on the error statistics of each source estimate, not on the
> number of speakers. Adding more speakers changes the difficulty of the separation problem, but it
> does not change or challenge the proposed early-exit mechanism itself.
>
> The community overwhelmingly uses 2-speaker mixtures (e.g., WSJ0-2mix, Libri2Mix) as the
> canonical setting for analyzing architectural contributions. Extending to 3+ speakers primarily tests
> the raw capacity of the separator – not the mechanism for probabilistic early exit.
> We believe our contributions are fully exercised and validated in the 2-speaker setting and have pri-
> oritized including the denoising setting over a 3+ speaker variation. Running the same experiments
> with more speakers would test scaling, not the proposed method.
> For these reasons, we do not currently prioritize >2 speaker experiments in the rebuttal phase as it
> would not substantially strengthen the evaluation of our method; however, we view multi-speaker
> extensions as a promising future direction and will discuss it in the revised document.
>
> **Question 3: Could you consider more perceptual metric for evaluation such as STOI and
> PESQ?**
>
> The results we present for speech enhancement in Table 3 of the paper include STOI and
> PESQ comparisons because some other works do not report SI-SNRI/SDRi for speech enhancement
> tasks, leaving PESQ and STOI as the only viable points of comparison. As explained in our comment
> to reviewer 2GXJ under ("SNR-centric"), both STOI and PESQ are poorly suited to evaluating the
> quality of neural network-based speech separation systems, and since the reporting of SI-SNRi and
> SDRi is common for the speech separation datasets we consider, we do not believe their inclusion
> would significantly strengthen the paper.

---

> > ### Comment · Reviewer_9ZZr · 2025-11-26
> >
> > Thank you for addressing my concerns and for your valuable answer.

---

### Official Review · Reviewer_DGuZ · 2025-10-30

**Soundness:** 3
**Presentation:** 4
**Contribution:** 3
**Rating:** 8
**Confidence:** 4

**Summary:**

This paper proposes PRESS, a probabilistic early-exit framework for speech separation and enhancement networks that enables dynamic computation scaling. The method introduces a neural architecture with multiple exit points, each predicting a clean speech signal and an error variance parameterized by an inverse-gamma distribution to model uncertainty. Evaluated on standard benchmarks, PRESS achieves performance competitive with state-of-the-art static models while providing interpretable, SNR-based exit conditions that can be calibrated on validation data for efficient inference.

**Strengths:**

- The paper demonstrates high originality. While early-exit mechanisms and probabilistic deep learning are known concepts, their application to speech separation is novel. The formulation of a probabilistic model that jointly estimates the clean speech signal and its error variance, parameterized by an inverse-gamma distribution, allows the authors to derive a principled early-exit condition. This is different from common heuristics like entropy or difference between successive blocks. The proposed "unified exit-SNR" condition, which combines SNR, SNR improvement, and a loudness condition, is a clever solution to the problem of silent segments, further showcasing the depth of the original contribution.
- The proposed PRESS is technically sound, evidenced by rigorous experimental design and compelling results. The paper includes thorough ablations that validate core design choices (e.g., the Student-t likelihood vs. SI-SNR, joint permutation invariance). The models are evaluated on three standard benchmarks (WSJ0-2mix, Libri2Mix, DNS Challenge), demonstrating the generality of the approach for both separation and enhancement tasks. The results show that PRESS achieves state-of-the-art performance at its deepest exit points.
- The paper is very well-written and clearly structured. The motivation -- enabling dynamic computation for heterogeneous devices, is immediately established. The probabilistic framework is explained step-by-step, with a clear derivation of the likelihood and the subsequent SNR distributions, making the theoretical contribution accessible. The results are presented effectively with tables and informative plots, such as the performance-vs-compute graph (Figure 3) and the calibration curves (Figure 4), which directly illustrate the core advantages of the method. The appendix provides valuable supplementary material, including convergence analysis and implementation details.

**Weaknesses:**

- The ablation study in Table 1 shows that the proposed Student-t likelihood performs on par with the standard SI-SNR loss. While the probabilistic framework is elegant and enables the exit condition, the paper does not demonstrate that this formulation provides a training advantage. The core contribution of the probabilistic model is its utility for uncertainty-aware exiting, but if a model trained with standard SI-SNR and a heuristic exit condition (e.g., based on the norm of the change between layers) performed similarly, the necessity of the more complex probabilistic training objective would be questioned.
- The paper's most significant limitation is the modelling of a single, global variance parameter $\sigma^2$ per exit for the entire audio segment. It must process the entire segment (e.g., 4 seconds at training, variable at test) to compute a global variance before any exit decision can be made. This undermines the goal of "fine-grained dynamic compute-scaling" for live applications, as the compute savings are only realized at the segment level, not within a segment.
  - To truly meet its stated goal, the authors should explore a time-varying variance model, predicting a frame-level or chunk-level variance $\sigma_t^2$ at each exit point. The early-exit condition could then be evaluated on a rolling window (e.g., the past 500ms). This is a non-trivial but crucial extension that would make the system causal and demonstrate true fine-grained, adaptive computation.

**Questions:**

The paper shows the performance available at each exit point, but not the performance achieved when the probabilistic exit condition is actively used. What is the average computational cost (e.g., in GMAC/s) and the corresponding average SI-SNRi when processing the entire WSJ0-2mix test set with your proposed exit condition (e.g., with a target of t=20 dB and p=0.9)?

 In Equation 7, the unified exit condition is defined as the maximum of the individual complementary CDFs. Could you provide more intuition for why the maximum is the right operator here? Was there an ablation comparing this to other operators (e.g., a weighted average or a product of probabilities)?

---

> ### Author Response · Authors · 2025-11-25
>
> **Weakness 1: ("Simpler heuristics")**
>
> While a model trained to optimize a combination of SI-SNR
> and some heuristic utilization loss may be conceptually simpler than our method, it may lead to
> unintended side-effects when actually used for early-exit as also described in our answer to reviewer
> eFq7 under ("Missing comparisons"), something that our framework makes much more explicit by
> directly reasoning about the relationship of the signal and the error. As we also point out in our
> response to reviewer eFq7, it is further difficult to quantitatively compare our early-exit condition to
> another early-exit condition as they express different notions of when performance is "good enough".
> We will attempt to better qualitatively distinguish and motivate our methods in our revised paper.
>
> **Weakness 2: ("Global variance parameter")**
>
> We agree with the reviewer’s assessment that a
> central limitation of the model, as presented, is its single, global variance parameter σ2. To ad-
> dress this, we have constructed a variant of PRESS that operates on blocks in the decoder. The
> blocked PRESS encodes the signal in an unchanged manner, but parameterizes the inverse-gamma
> distributions in blocks of a fixed size.
>
> **Question 1: "The paper shows the performance available at each exit point, but not the per-
> formance achieved when the probabilistic exit condition is actively used. What is the average
> computational cost (e.g., in GMAC/s) and the corresponding average SI-SNRi when processing
> the entire WSJ0-2mix test set with your proposed exit condition (e.g., with a target of t=20 dB
> and p=0.9)?"**
>
> Computing the SI-SNRi per GMAC/s forms a Pareto curve (as a function of either t
> or p, which would yield similar behavior of monotonically increasing both the average SI-SNRi and
> the GMAC/s), which we will include in figure 3.
>
> **Question 2: "In Equation 7, the unified exit condition is defined as the maximum of the indi-
> vidual complementary CDFs. Could you provide more intuition for why the maximum is the
> right operator here? Was there an ablation comparing this to other operators (e.g., a weighted
> average or a product of probabilities)?"**
>
> We use the maximum because each SNR-based distribution (SNR, SNRi, SNRref) represents an independent sufficient condition for exiting. The max
> operator is therefore the probabilistic equivalent of a logical OR so if any condition is confident that
> the target SNR has been reached, further computation is unnecessary.
> Other operators change the semantics in undesirable ways. A (weighted) average or a product effec-
> tively imposes an AND-like requirement, making the exit rule overly conservative and reintroducing
> the very failure modes (e.g., silence, low interference) that the unified condition is designed to avoid.
> Because these operators correspond to different decision logics, rather than plausible variants of our
> intended rule, ablations are not informative. Thus, we believe the maximum is the appropriate and
> principled choice.

---

### Official Review · Reviewer_2GXJ · 2025-10-31

**Soundness:** 2
**Presentation:** 2
**Contribution:** 2
**Rating:** 4
**Confidence:** 3

**Summary:**

The paper proposes PRESS, a technique for performing speech separation and enhancement that can dynamically scale to various compute budgets. They achieve this by designing network architectures equipped with “early exits”. These exits are driven by probabilistic SNR (&SNRis) based criterion, and the network stops at the first depth where the probability of SNR > a set target. Results on WSJ0-2mix, Libri2Mix, and DNS Challenge datasets were presented.

**Strengths:**

1. The problem is well motivated. Having the ability to adaptively exit from the computational chain is especially useful on compute and energy-constrained devices such as VR headsets, hearbles, and smart speakers. So studying this seems a useful task for practical deployments.

2. Basing the exit criterion on SNR-type variants seems intuitive and logical, given that they have easy-to-tune parameters.

3. Experimental results at the deepest exit are competitive with shown baselines. Abaltions are helpful.

**Weaknesses:**

1. Novelty: While the paper tries to solve a practical problem, it seemed like an incremental improvement to previous works (e.g., SepReformer) and did not come across as novel enough.
The architecture largely follows SepReformer (early split, transformer stack, CLA and other layers), with the main additions being extra exit heads and a standard likelihood derived from conjugate priors. Also, early exit itself is a known idea, and turning the SNR threshold into an exit confidence feels like a natural application of that likelihood.

2. Empirical results: While the depth-wise exits show incremental improvements (Table 2), they are not surprising. Moreover, why not simply use SepReformer Tiny/Small, which has fewer parameters yet achieves similar (and in some cases better) performance? Early exits don’t really reduce the on-device model size anyway; the full network (with the exit heads) must still be stored, so gains are mainly runtime latency or energy saving on easy inputs, not footprint. But even the #GMAC/s reported don’t clearly beat other baselines. So it’s unclear to me what the concrete gains of the proposed approach are relative to baselines.

3. Unified early exit: The reasoning behind using a combined SNR exit criterion (Eq. 11)  when at least one of them exceeds the set threshold seems heuristic and is not quite convincing. What if you make all three exceed the threshold? Also, I wonder if the thresholds are always triggered mostly by a particular exit. Ablations corresponding to these are missing.

4. Conjugate prior:  The choice of an inverse-Gamma prior on $\sigma^2$ seems to be made mainly for conjugacy and convenience, and there’s little justification for why that is the case. It would help to compare results against a simpler case, such as a Gaussian with both mean and sigma learned as point estimates, removing the need for marginalization.

5. SNR-centric: The paper relies heavily on the SNR-style variants for both the exit criterion and evaluations. It would help the reader to see whether these SNR-based exit choices are a key design choice. Also, reporting with other perceptual metrics, such as PESQ, STOI would help make better judgment of the method.

6. Writing: The paper is somewhat of a hard read; Section 3.1, in particular, feels under-motivated and confusing. For instance $x$ is not defined in Eq. 1, and similarly for T in Eq. 3 (the signal length). At line 200, in Eq. (3), doesn’t the second-to-last term actually penalize over-estimating the variance?

Minor comments:
L234: correction at p(SNR(x, \hat{x}) \geq t) \geq p to Pr(.)

**Questions:**

Questions:
I would like the authors to comment on these:
1. Eqs. 8 and 9 approximate SNR by 1 + $\lambda/T$ where $\lambda$ is the non-centrality parameter when T tends to infinity. How does this behave for shorter time windows and how does this hold in real scenarios?
2. Other than simplifications, what is the authors’ rationale behind using the inverse-gamma as a conjugate prior? Anything specific about the variance that motivates the prior? Does restricting the noise model to the Normal-Inverse-Gamma family limit performance? Because once the classes are restricted, the performance may be bounded? Is this the case?
3. A stretch, but can we see this depth-quality as analogous to the sampling time-quality in diffusion models? i.e, can diffusion models naturally achieve this dynamism simply by varying the number of sampling steps? Can the authors draw any parallels between the two?

---

> ### Author Response · Authors · 2025-11-25
>
> **Weakness 1: ("Novelty")**
>
> We argue that the contribution of the paper is substantial. PRESS
> in architecture deviates considerably from SepReformer whereas our likelihood based objective is
> also novel in the context of audio processing. Architecturally, SepReformer as-is is not suitable
> for early-exit because it downsamples the audio signal dramatically for each stage in its U-Net
> architecture and its early "reconstructions" only predict absolute STFT magnitudes, not an actual
> waveform. We proposed PRESS specifically to maintain or exceed SepReformer performance while
> being suitable for early-exiting (i.e. having strong early reconstructions). We will add this crucial
> point of motivation to the paper, which is complementd by our new results showing that an early-
> exiting PRESS model matches or exceeds the per-exit performance of same-size single-exit model.
> Specifically, the PRESS architecture substantially deviates from SepReformer by
>
> 1. Not relying on a U-net architecture.
> 2. Reconstructing all exit points in full resolution as opposed to reconstructing only the absolute
> STFT magnitudes at intermediate exits in terms of down-sampled representations of the signals.
> 3. Not relying on self-attention transformers but linear RNNs for efficient long range context learning, as self-attention would be prohibitively expensive without internal downsampling as in SepReformer.
> 4. Train using a new loss specification based on the proposed student’s t likelihood, which can be
> interpreted as a compound distribution of a normal distribution with unknown variance specified by
> the inverse-gamma distribution.
> Importantly, we highlight how this new likelihood specification enables uncertainty awareness and
> naturally admits to deriving early-exit conditions from the learned characterization of the uncertainty
> while at the same time naturally weighting the influence of the different exit points when learning
> the model by simply minimizing the sum of all exit points’ likelihood. Thereby, the developed
> framework allows a simple unified learning framework that naturally combines multiple exits in the
> modeling while our proposed Student’s t-likelihood can also be viewed from the perspective of a
> normal-inverse-gamma compound distribution not only producing variance estimates but characterizing the uncertainty of the estimated variances that naturally can be used to produce early-exit conditions. While the Student’s t-based likelihood is well known and the relation between the Student’s
> t-likelihood and the normal-inverse-gamma compound distribution is as well this has not previously
> to the best of our knowledge been explored in a deep learning context to create uncertainty-aware
> modeling nor explored in the context of early-exit architectures where we demonstrate it naturally
> weights the influence of the different early exits and produce actionable exit conditions.
>
> **Weakness 2: ("Novelty and empirical results")**
>
> As described above SepReformer is not suitable
> for early exit as-is. While a large part of our motivation and evaluations center on saving compute,
> we would also like to highlight that our method provides a principled and more flexible method for
> early-exit decision-making than prior works which are typically based on ad-hoc trade-offs between
> reconstruction and utilization which are baked into the model training, whereas our models can be
> used with any target SNR levels.
> The gains of the proposed approach is an end-to-end learning framework that naturally accommodates and weights the influence of multiple exits while producing uncertainty aware modeling that
> results in actionable early exit conditions at inference time without additional modeling. Importantly our approach is the first to explore early-exit in a SOTA-level model for speech separation and
> demonstrate that reconstruction performance need not be impaired by training with early exits.
> In addition to these points, our new architecture simplifications have improved performance to place
> our model significantly above SepReformer performance per compute.
>
> **Weakness 3: ("Unified early exit")**
>
> We argue that the unified early exit is a strength of our
> approach highlighting how multiple exit conditions can be derived and combined to explicitly accommodate accurate decision making when producing the early exits.
> "What if you make all three exceed the threshold?": could the reviewer please clarify what was
> meant by this? Our method early exits if at least one (or more) conditions are met.
>
> "Also, I wonder if the thresholds are always triggered mostly by a particular exit": this is already
> partially shown in our current results, as (1) our models provide non-trivial performance improvements per-exit, and (2) our SNR distributions are probabilistically post-calibrated, and (3) figures 1
> and 6 show an example where the exit-SNR exceeds the threshold at very different exit points. We
> will however consider how to show this more clearly and holistically.

---

> ### Author Response · Authors · 2025-11-25
>
> **Weakness 4: ("Conjugate prior")**
>
> Our interest is training the model using a simple likelihood
> that naturally accommodates uncertainty awareness while corresponding to a loss closely aligned
> with existing SI-SNR based training. We describe this in appendix G highlighting a close relationship between a scale-invariant Student’s t-likelihood and SI-SNR, as well as how the present
> Student’s t-likelihood closely relates to the existing log-RMS error that has been found to perform
> comparably to SI-SDR, see reference (Heitkaemper et al., 2020) in appendix G. A similar point
> can be made by noticing that the normal-inverse-gamma induces much more heavy-tailedness in the
> error distribution than a plain normal distribution would. We observe in our new ablations that the
> t-likelihood obtains better reconstruction performance than a normal likelihood when evaluated in
> terms of SI-SNR.
>
> **Weakness 5: ("SNR-centric")**
>
> While we do not report PESQ or STOI for the speech separation
> task, we do report STOI and WB-PESQ in Table 3 for the speech enhancement task to facilitate
> comparison with other works who only report these, while in speech separation most works rely on
> SI-SNR/SI-SDR and SDR (improvement) for comparison.
> Perceptual metrics for audio quality are known to be limited and often unreliable outside their original design conditions (e.g., [3]). Given that our work focuses specifically on early-exit behavior
> derived from predictive SNR distributions, we use SNR-based losses and metrics, which remain the
> standard in the field and directly aligned with our probabilistic formulation. There are also technical
> reasons for preferring SNR-based objectives: unlike the default PESQ and STOI formulation, they
> are fully differentiable and therefore compatible with end-to-end training. While designing architectures optimized directly for perceptual metrics is a very interesting direction for future work, we still
> believe this is an open problem in the deep learning for audio domain and not yet standard practice.
> As such, we believe the focus on SNR-based metrics is both appropriate and sufficient for the contributions of this paper, but we will include a discussion and justification for the SNR-focus in the
> revised paper.
>
>
> [3] Espejo, I. L., Edraki, A., Chan, W.-Y., Tan, Z.-H., & Jensen, J. (2023). On the Deficiency of
> Intelligibility Metrics as Proxies for Subjective Intelligibility. Speech Communication, 150, 9-22.
> https://doi.org/10.1016/j.specom.2023.04.001
>
>
> **Weakness 6: ("Writing")**
>
> We apologize for any unclear notation, and we will in the revised
> manuscript include a paragraph describing all the relevant notation used upfront.
>
> **Question 1: "Eqs. 8 and 9 approximate SNR by 1 + λ/T where λ is the non-centrality parameter when T tends to infinity. How does this behave for shorter time windows and how does
> this hold in real scenarios?"**
>
> We investigate the validity of the approximation as function of T in
> Figure 5 of supplementary section A. We here find that for the parameters corresponding to the average model predictions seen during training the Kolmogorov-Smirnov (KS) test statistics are below
> 1% for T = 2000 supporting the validity of the approximation in the considered real scenarios. We
> will update this to also include such a simulation using the parameters predicted by a real model on
> test data.
>
> **Question 2: ("Other than simplifications, what is the authors rationale behind using the inverse-
> gamma as a conjugate prior? Anything specific about the variance that motivates the prior?
> Does restricting the noise model to the Normal-Inverse-Gamma family limit performance? Because once the classes are restricted, the performance may be bounded? Is this the case?")**
>
> See our response above to Weakness ("Conjugate prior").
> In summary, the central goal is to
> achieve a likelihood based training framework that works well which the Student’s t-based likelihood provides as discussed in Appendix section G. Notably, this likelihood can be defined as the
> normal-inverse-gamma compound distribution and this relation is in turn explored to provide the
> uncertainty estimates of the variance used for knowing when to quit.

---

> ### Author Response · Authors · 2025-11-25
>
> **Question 3: "A stretch, but can we see this depth-quality as analogous to the sampling time-
> quality in diffusion models? i.e, can diffusion models naturally achieve this dynamism sim-
> ply by varying the number of sampling steps? Can the authors draw any parallels between
> the two?"**
>
> Conceptually yes, as we discuss in our related works section lines 126-133, diffusion
> models also trade off computation for successively increased quality, but notoriously struggle with
> few-step performance and even when using few-step methods may also be limited by the need to
> reconstruct the audio waveform with every forward pass (our method does not need to re-encode
> the waveform for each level), or requiring a two-stage training for latent diffusion. The relationship
> is even more apparent if we impose that the same architecture is applied recursively at each step as
> in diffusion models as opposed to presently considering separate decoder blocks for each exit. In
> turn, such recursive architectures would also allow for arbitrary exit depths, as briefly mentioned
> in our conclusion. We will add some extra clarifications on distinctions from and similarities to
> diffusion/flow models to our related works section.

---

### Official Review · Reviewer_eFq7 · 2025-11-02

**Soundness:** 3
**Presentation:** 3
**Contribution:** 2
**Rating:** 4
**Confidence:** 4

**Summary:**

PRESS introduces an early-exit architecture for speech separation that enables dynamic compute scaling by allowing the network to stop processing at different depths.
This is achieved by a probabilistic framework that models the predicted SNR distributions, exiting when the predicted SNR , SNRi or a loudness condition exceeds a target threshold with specified confidence.
The proposed method offers competitive results with state-of-the-art models on WSJ0-2mix, Libri2Mix, and DNS2020 while enabling compute reduction.

**Strengths:**

This paper is the first to apply uncertainty-aware probabilistic modeling specifically for early-exit decisions in speech separation.
One great advantage of the proposed method is some interpretability in the chosen early exit strategy.
The paper has strong empirical results with respect to the SotA non early exiting models on 3 different datasets including DNS challenge which features speech enhancement.

**Weaknesses:**

This work misses a crucial comparison with another work which also explored early exit for speech separation already in 2023:

Bralios, D., Tzinis, E., Wichern, G., Smaragdis, P., & Le Roux, J. (2023, June). Latent iterative refinement for modular source separation. In ICASSP 2023-2023 IEEE International Conference on Acoustics, Speech and Signal Processing (ICASSP) (pp. 1-5). IEEE.

This work goes into the direction outlined by the authors in the future possible works where instead of using different blocks one uses iterative models.
Another key different between the two works is that the early exiting mechanism is fundamentally different as PRESS relies on estimated SNR.
However I feel that this work should be discussed and maybe compared against e.g. by having a baseline system adopting shared parameters and gating-based stopping.

Another work is mentioned which explored early exit for speech separation:

Chen, S., Wu, Y., Chen, Z., Yoshioka, T., Liu, S., Li, J., & Yu, X. (2021, June). Don’t shoot butterfly with rifles: Multi-channel continuous speech separation with early exit transformer. In ICASSP 2021-2021 IEEE International Conference on Acoustics, Speech and Signal Processing (ICASSP) (pp. 6139-6143). IEEE.

I think another drawback of this paper is that the proposed early exit strategy is not compared with the strategy adopted in this latter paper. Does proposed probabilistic SNR work better than Euclidean norm difference ?
Which exit criterion transfers better across domains? is PRESS estimation robust ?
Right now these questions are unsolved and it is unclear how significant is the proposed method compared to the previously proposed two.

Also the paper lacks other ablations to justify the proposed probabilistic early exiting framework.
For example what if instead of the probabilistic framework I use simply a supervised predictive head to estimate the SNR ?
Or what if only p(SNR > t) is used as an early exit condition ? Figures 1and 6 compare the different strategies but there are no tables.

Unfortunately due to the lack of these experiments the paper as it is right now is not suitable for publication. While the proposed strategy seems very strong it is unclear if it is more effective than others (and possibly simpler) strategies.

**Questions:**

I ve put questions in the weaknesses section.

---

> ### Author Response · Authors · 2025-11-25
>
> **Weakness 1: ("Missing comparisons")**
>
> We thank the reviewer for bringing the prior work [1] to
> our attention, which should have been referenced. The two works pointed out by the reviewer are
> good examples of the two ways that prior works tend to set up early exit:
>
> In short, the method of early-exiting in [1] is based on the model learning a trade-off induced by
> a reconstruction loss component and a network utilization loss component. Crucially, the trade-off
> between reconstruction and utilization is fixed at training time through the functional form of the
> utilization loss and the relative weighting of reconstruction and utilization in the total loss, rendering
> these methods difficult to use in practice, requiring retraining if the target level must change. Additionally, when SI-SNR is used as the reconstruction loss, the utilization implicitly becomes loudness
> sensitive when the loudness within the audio clip varies significantly, as quieter parts will contribute
> less to the signal and noise sum in the reconstruction. While this is not necessarily a bug, our method
> explicitly ties the signal and the estimated error, making this effect much clearer. In our new results using a blocked likelihood, only local parts of the signal are used with local error estimates making
> our method loudness insensitive across blocks.
>
> In [2] the early-exit mechanism is based on convergence of the reconstruction (Euclidean distance
> between consecutive early-exits), which does not guarantee any particular level of performance,
> making it unclear what the actual performance goal is, and is also implicitly loudness sensitive due
> to the use of Euclidean distance. Even if SI-SNR between consecutive exits were used, it would still
> be loudness sensitive in the same sense that the use of SI-SNR in [1] is. In contrast, our work is the
> first to ground the early-exit mechanism in an explicit and directly interpretable SNR condition, and
> the target SNR condition can be readily changed at inference time by simply evaluating the proposed
> CDF at a different target level.
>
> In our revised paper we will add a new paragraph to the introduction recapping the current state
> of early-exit methods for speech separation/enhancement including the above points to improve
> readability and make the motivation for our work clearer.
>
> **Question 1: ("Does proposed probabilistic SNR work better than Euclidean norm difference?")**
>
> It is generally unclear how the exit conditions in [1], [2] and our work could be compared
> on a scale of "worse-to-better", as they express different beliefs about when the reconstruction is
> "good enough" and further improvement is wasteful. With our work we make the case that a more
> explicit and interpretable method such as ours which is easier to reason about is preferable to implicit ones which may have unintended behaviors (e.g. implicit loudness sensitivity) and require
> ad-hoc utilization loss design. Another advantage of our work is that we have systematically demonstrated that our method does not impair reconstruction performance, an important baseline which
> prior works do not have.
>
> **Question 2: ("Which exit criterion transfers better across domains?")**
>
> While generalization /
> out-of-distribution evaluation would be interesting, we consider it outside the scope of this work for
> now. We do highlight that our method is amenable to post-calibration, which could be extended past
> our simple method with e.g. distributional conformal prediction to give calibration guarantees even
> in out-of-distribution evaluations, which it would be unclear how to approach with prior methods.
>
>
>
> [1] Bralios, Dimitrios, et al. "Latent iterative refinement for modular source separation." ICASSP
> 2023-2023 IEEE International Conference on Acoustics, Speech and Signal Processing (ICASSP).
> IEEE, 2023.
>
> [2] Chen, Sanyuan, et al. "Dont shoot butterfly with rifles: Multi-channel continuous speech separa-
> tion with early exit transformer." ICASSP 2021-2021 IEEE International Conference on Acoustics,
> Speech and Signal Processing (ICASSP). IEEE, 2021.

---

> ### Author Response · Authors · 2025-11-25
>
> **Question 3: ("Is PRESS estimation robust?")**
>
> With our latest full-length finetuning experiments
> we find that PRESS IS robust without any extra post-calibration, identifying our prior train/val/test
> mismatch as an out-of-distribution issue. We evaluate this in terms of the expected calibration
> error (ECE), and will also include new metrics for quantifying distributional regression methods,
> including the continuous ranked probability score (CRPS) and probability integral transform (PIT)
> ECE to better quantify the performance of the predicted SNR distributions.
>
> **Question 4: ("What if instead of the probabilistic framework I use simply a supervised predic-
> tive head to estimate the SNR? Or what if only p(SNR > t) is used as an early exit condition?")**
>
> We have added new ablations using a Gaussian-only likelihood (resulting in worse performance than
> our proposed t-likelihood), which we hope partially answers the reviewer’s first part of the question
> as it represents letting the variance be a simple point estimate made by the network (analogous to a
> prediction head estimating the SNR, as one can then compute an estimated SNR using the predicted
> signal power and estimated error), but as explained in response to ("Missing comparisons"), it is still
> not clear how to quantitatively compare our exit conditions to a model estimating only the (SI-)SNR.
> In general it is not sufficient to consider only SNR or SNRi if the method of early exit should take
> advantage of silence, since as we argue on lines 237-241, if a substantial part of the audio is silence,
> the SNR goes to zero and essentially become insensitive to the loudness of the noise signal. We
> show this effect qualitatively in figures 1 and 6.

---

> > ### Comment · Reviewer_eFq7 · 2025-11-26
> > **Thank you for addressing my concerns**
> >
> > I have raised my score to 6.

---

### Author Response · Authors · 2025-11-25

We thank the reviewers for their thorough and constructive feedback. We have made several improvements to the paper since submission that we list here:

- We have modified the architecture to consist entirely of our proposed linear RNN blocks
and speaker attention blocks, both simplifying the architecture and improving performance
per GMAC/s. We have also found that some of our architectural choices such as Snake activations
, patch encoder/decoder and ShiftNorm no longer offer any advantage over more
conventional alternatives such as GELU and convolutional encoder/decoder, with the simplified
 architecture, and we have therefore switched to these instead.
Crucially, these
changes greatly simplify the exposition of the model architecture in the paper, and leads to
performance per GMAC/s improvements across all datasets. This model converges somewhat
more slowly, and we have found that extending training time to 6M steps is necessary
to reach fully converged performance.

- We have included results on the noisy 2-speaker WHAM! and reverberant WHAMR!
datasets for more realistic audio data.

- We have trained models that predict the variance of small chunks of audio rather than
globally, to investigate the effect of more time-local early-exit. We present results for
models trained with audio block sizes of T = 500, 1000, 2000, 4000 samples, as well as
our non-blocked models. In the decoding process, the model predicts the variance for
chunks of the sequence as opposed to one global variance for the entire sequence.

- We have included new ablations that investigate (1) reconstruction performance of our early-
exit models against same-size single-exit models and find that we now match the per-exit
performance against these, and (2) a Gaussian-only likelihood, which results in worse per-
formance than our t-likelihood.

- We have done further experiments regarding calibration and found that uncalibrated error
distributions can be remedied by finetuning models on full-length training data, i.e. continuing
training (in our experiments for 200K steps, ca. 3% of base training time) with
full-length sequences rather than cropping audio to 4 second clips as in the bulk of training.
This shows that the uncalibrated error distributions seen on validation and test data stem
from an out-of-distribution issue (train on 4 seconds, evaluate on e.g. 15 second clips). We
ablate that this effect is indeed caused by the variable-length training data by checking that
finetuning on 200K steps after the base training while still cropping to 4 seconds does not
lead to improvement. While this obviates the need for a post-calibration step in our case,
we will still include our post-calibration method in the appendix as it may be useful on
datasets with train/test discrepancies.

We will include these new results in a revision of our paper in the coming days.

---

> ### Author Response · Authors · 2025-12-03
>
> We once again thank the reviewers for their invaluable feedback. We have now uploaded our latest revision to the paper, which covers (among others) the following key points from the original reviews:
>
> * New results, as explained in our original comment.
> * New text in the introduction motivating our method.
> * New evaluations of calibratedness in terms of probability integral transform and continuous ranked probability score.
> * New evaluations of early-exit performance in figures 3 and 5, which show that our early-exit conditions can beat static single-exit models both in terms of SI-SNRi and in terms of exit-SNR regret, closely approaching oracle performance.

---

### Meta-Review · Area_Chair_r84D · 2026-01-05

**Summary:**

Multiple reviewers mentioned that there should be comparisons to existing early-exit strategies.Reviewer eFq7 highlights the lack of comparison against other early exit works (Bralios et al., 2023; Chen et al., 2021). Similarly, Reviewer DGuZ also questioned the necessity of the complex probabilistic training. Additional weaknesses include the marginal performance improvements on SOTA results [2GXJ, 9ZZr, fRD5], the architectural novelty being incremental [2GXJ], and the lack of justification for the inverse-Gamma prior [2GXJ]. Also, some miscellaneous comments regarding writing and clarity.

**Reviewer Concerns:**

From the AC’s perspective, it seems that all the reviewers' concerns were addressed. Given that the authors have included additional comparisons in the results and clarified the motivation.

**Reviewer Scores:**

Most of the reviewers' scores are positive, and the AC does not see that the rebuttal would lower their score. Reviewer eFq7 has commented that they would raise their score to a 6. Reviewer 2GXJ’s questions seem adequately addressed; hence, the score would probably go up.

---

### Decision · Program_Chairs · 2026-01-26

Accept (Poster)